# Quantification of climate change impact on dam failure risk under hydrological scenarios: a case study from a Spanish dam

Javier Fluixá-Sanmartín[1], Adrián Morales-Torres[2], Ignacio Escuder-Bueno[2,3], and
Javier Paredes-Arquiola[3]

[1]Centre de Recherche sur l'Environnement Alpin (CREALP), Sion, 1951, Switzerland
[2]iPresas Risk Analysis, Valencia, 46023, Spain
[3]Research Institute of Water and Environmental Engineering (IIAMA), Universitat Politècnica de València (UPV), Valencia, 46022, Spain

**Correspondence:** Javier Fluixá-Sanmartín (javier.fluixa@crealp.vs.ch)

**Abstract.** Dam safety is increasingly subjected to the influence of climate change. Its impacts must be assessed through the integration of the various effects acting on each aspect, considering their interdependencies, rather than by a simple accumulation of separate impacts. This serves as a dam safety management supporting tool to assess the vulnerability of the dam to climate change and to define adaptation strategies under an evolutive dam failure risk management framework.

This article presents a comprehensive quantitative assessment of the impacts of climate change on the safety of a Spanish dam under hydrological scenarios, integrating the various projected effects acting on each component of the risk, from the input hydrology to the consequences of the outflow hydrograph. In particular, the results of 21 regional climate models encompassing three Representative Concentration Pathways (RCP2.6, RCP4.5 and RCP8.5) have been used to calculate the risk evolution of the dam until the end of the 21st century. Results show a progressive deterioration of the dam failure risk, for most of the cases contemplated, especially for the RCP2.6 and RCP4.5 scenarios. Moreover, the individual analysis of each risk component shows that the alteration of the expected inflows has the greater influence on the final risk. The approach followed in this paper can serve as a useful guidebook for dam owners and dam safety practitioners in the analysis of other study cases.

## 1 Introduction

Dams are critical infrastructures whose associated failure risk must be properly managed in a continuous and updated process (Fluixá-Sanmartín et al., 2018). When assessing their safety levels, most dam risk assessments in the past assumed a stationary condition in the variability of climate phenomena. However, climate change is likely to affect the different factors driving dam failure risks (USBR, 2014). The assumptions of stationary climatic baselines are no longer appropriate for long-term dam safety adaptation and decision-making support (USBR, 2016). Therefore, the way risk analyses are envisaged on the long term has to be revisited in order to incorporate the new climate change scenarios.

In this context, some efforts have been done in the evaluation of climate change impacts on dam safety surveillance (OFEV, 2014; USACE, 2014; USBR, 2014, 2016). However, the assessment of these impacts is usually applied separately and tend to focus on specific aspects such as the hydrological loads (Bahls and Holman, 2014; Chernet et al., 2014; Novembre et al., 2015) relegating or ignoring other aspects.

The global effect of climate change on dam safety must be quantitatively assessed through the integration of the various projected effects acting on each aspect. In Fluixá-Sanmartín et al. (2018), a dam safety management supporting tool is defined to assess projected climate change impacts based on the risk analysis approach where all the variables concerning dam safety and their interdependencies could be included in a comprehensive way. In this context, risk analysis is a useful approach encompassing traditional and state-of-the-art methodologies to manage dam safety in an accountable and comprehensive way
(Bowles, 2000; Serrano-Lombillo et al., 2013) that represents a useful basis on which such assessments can be structured. With this quantitative information, long-term investments can be planned more efficiently taking into account the potential evolution with time of risk and of the efficiency of measures.

In this work the authors seek a comprehensive quantitative assessment of the climate change impacts on the failure risk of a Spanish dam. The key innovative aspect of this methodology is the use of very different models and data sources, and
their combination for the assessment of the overall effect of climate change in the resulting dam safety risk. The analysis has been elaborated under hydrological scenarios, where the floods are the main loads to which the dam is subjected. In order to decompose such impacts on the different risk aspects, a risk analysis scheme has been adopted. First, the methodological approach proposed is presented. Then the study case of the Santa Teresa dam to which the methodology will be applied is described. The different data sources and existing models employed on this study are presented. Using this information, the
methodology is applied to the study case, explaining the treatment of raw climate projections, the elaboration of auxiliary models and the adaptation of the risk model components. Finally, the output risks are presented and the resulting effects on the dam safety analysed.

## 2   Methodology

This section describes the methodology proposed in this paper for the calculation of climate changes impacts on the safety of
dams. The goal is to analyse its effects on the different dam failure risk components involved. It is worth noting that, within the context of dam safety, failure risk can be defined as the combination of three concepts: what can happen (dam failure), how likely it is to happen (failure probability), and what its consequences are (failure consequences) (Kaplan, 1997). Risk is obtained through the following formula:

$$Risk = \sum_{e} p(e) \cdot p(f|e) \cdot C(f|e) \qquad (1)$$

where the risk is expressed in consequences/year (social or economic), the summation is defined for all events $e$ under study, $p(e)$ is the probability of an event, $p(f|e)$ is the probability of failure due to event $e$ and $C(f|e)$ are the consequences produced as a result of each failure $f$ and event $e$.

As stated in Fluixá-Sanmartín et al. (2018), changes in climate such as variations in extreme temperatures or frequency of heavy precipitation events (IPCC, 2012; Walsh et al., 2014) are likely to affect the different risk components driving dam failure. Hence, the proposed methodology intends to establish a framework for the evaluation of projected climate change impacts on dam safety attending to both climatic and non-climatic drivers. This is based on the risk analysis approach where the effects on all the variables concerning dam safety – from the hydrological loads to the consequences of failure – and their interdependencies are evaluated jointly. The cornerstone of the methodology is the application of a dam risk modelling approach which encompasses the information issued from different models and data sources.

Moreover, since climate change is a non-stationary process, it is expected that its effects will change with time. Therefore, it is not only important to assess the global impact of climate change on the dam failure risk but also how this risk is expected to evolve with time. For this purpose, the methodology should be applied on one hand to the present situation (to which the future results will be compared) and on the other hand to different time horizons in the future. Given that the climate projections used in this study include results until the end of the 21$^{st}$ century, the following four different periods are proposed in this study:

- **Historical:** 1970-2005. It corresponds to the period for which hydro-meteorological observations are available, as well as to the reference historical period of the climate projections (cf. Sect. 4.2). This allows performing the downscaling of the climate projections. Such period will be referred as Base Case.

- **Period 1:** 2010-2039.

- **Period 2:** 2040-2069.

- **Period 3:** 2070-2099.

The methodology proposed is based on the following main steps. A synthetic scheme of this methodology is presented in Fig. 1.

(a) **Extraction and correction of climate projections.** First, the raw climate projections issued from the available climate models must be bias corrected using the climate observations. Assessing the impacts of climate change on future runoff generation and on water resources availability require high-resolution climate scenarios. Global Climate Models (GCM) provide valuable prediction information but at a spatial resolution too coarse (around 1 000 by 1 000 km) to be directly used for modelling the hydrological processes at the required scale (Akhtar et al., 2008; Fujihara et al., 2008; Orlowsky et al., 2008). Therefore, downscaling is required to describe the consequences of climate change, which can be done using empirical-statistical downscaling or dynamical downscaling by means of regional climate models (RCMs). RCMs are commonly used in regional studies of climate projection and climate change impacts to downscale GCM simulations (Gao et al., 2006; Gu et al., 2012; Yira et al., 2017). They use the GCM outputs as lateral boundary conditions and thus their results depend to some extend on its driving GCM (Benestad, 2016). However, the meteorological projections issued from RCMs are usually biased and hence need to be post processed before being used for climate impact assessment (Gudmundsson et al., 2012).

(b) **Hydrological modelling.** Then, a hydrological model is set up based on the physical characteristics of the basin and on the hydro-meteorological observations. On one hand, such model allows to perform the simulation of the system of water resources management to obtain the relation between previous pool level and probability at the reservoir, at the present situation and for future scenarios. On the other hand, the hydrological model is also used for the calculation of the flood hydrographs arriving into the reservoir.

(c) **Risk modelling.** The quantitative assessment of climate change impacts on dam failure risk is conducted using a quantitative risk model of the dam. As explained, such models are commonly used to inform dam safety management and they integrate and connect most variables concerning dam failure risk to analyse the different ways in which a dam can fail (failure modes) resulting from a loading event, calculating their probabilities and consequences (Ardiles et al., 2011; Serrano-Lombillo et al., 2011, 2012a, b; SPANCOLD, 2012). The model must be adapted following the effects of climate

change on each of the risk components (Fluixá-Sanmartín et al., 2018).

  (d) **Correction of resulting risks.** In order to consistently assess and compare modelled risks, a change signal correction (likewise the delta change approach) must be applied to the results by scaling the outputs based on the difference between climate model and Base Case risks for the historical reference period. This correction is computed as the relative variation between raw risk output for a future scenario and risk of its corresponding historical reference period. Then, the future

scenario risk is adjusted by multiplying this delta to the Base Case risk.

## 3   Study case

The Santa Teresa dam is located in the upper part of the Tormes River, in the Province of Salamanca (Spain), and is managed by the Duero River Basin Authority. The Santa Teresa reservoir is bounded by the Santa Teresa dam and a smaller auxiliary dike. The Santa Teresa dam is a concrete gravity dam built in 1960 and has a height of 60 m with its crest level at 887.20 m

a.s.l. and a length of 517 m. It is equipped with a spillway regulated by five gates capable of relieving, altogether, 2050 $m^3$/s, as well as with two bottom outlets with a release capacity of 88 $m^3$/s each. The auxiliary dike is a 165 m long and 15 m high concrete gravity dam with its crest level at 886.90 m a.s.l.

    The Santa Teresa reservoir has a capacity of 496 $hm^3$ at its normal operating level (885.70 m a.s.l.). The catchment that pours into the reservoir has a total surface of 1853 $km^2$ and is part of the Tormes Water Exploitation System, being the Santa

Teresa reservoir the first and uppermost infrastructure of the basin to regulate the Tormes River (Fig. 2). The main uses for the Santa Teresa dam-reservoir system are hydropower production, flood protection, irrigation and water supply to the demands located between the Santa Teresa and the Almendra dams, including Salamanca city.

    A risk analysis was already applied to the Santa Teresa dam in a previous study (Ardiles et al., 2011; Morales-Torres et al., 2016). Results from this analysis showed that, although the dam didn't require urgent correction measures, its risk was

important enough to be carefully monitored. Therefore, it is interesting to evaluate whether its risk is expected to evolve up to the point of requiring correction measures.

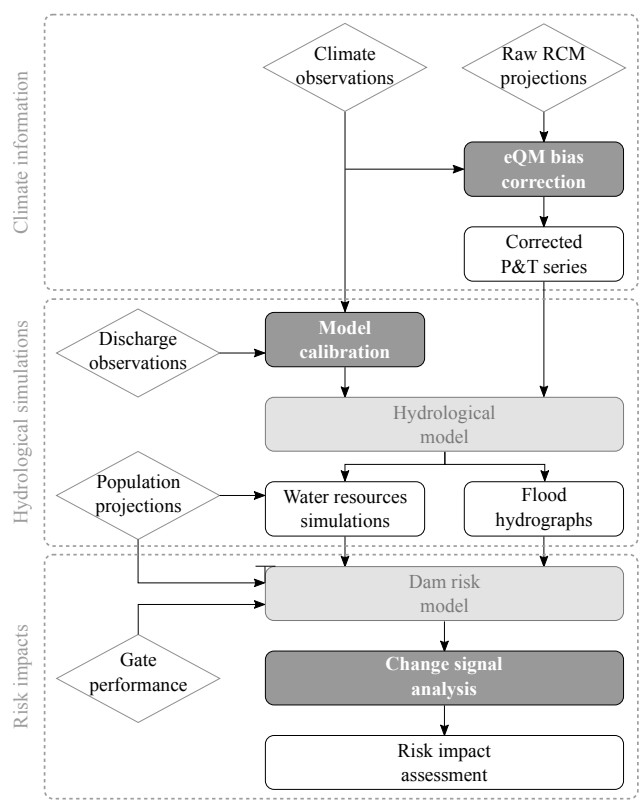

**Figure 1.** Workflow of the methodology followed to assess climate change impacts on dam failure risk.

## 4 Data and models

### 4.1 Hydrometeorological data

The meteorological inputs used for the definition of the present situation are based on the observed data collected by the Spanish
Meteorological Agency (AEMET). For this study, the Spain02 products have been employed. Spain02 is a series of high-resolution daily precipitation and mean temperature gridded datasets developed for peninsular Spain and the Balearic Islands.
A dense network of over 2 000 quality-controlled stations was selected from the AEMET and the Santander Meteorology
Group (University of Cantabria, 2019) in order to build the gridded products for the different dataset versions. The latest
version of the dataset (Spain02 v5) provides daily data from 1951 to 2015 in a 0.1º interpolated regular grid (Herrera et al.,
2016; Kotlarski et al., 2017). The full dataset is available at the AEMET climate services portal (AEMET, 2019).

For the calibration of the hydrometeorological model, the daily historical discharge records at nine different stations within
the catchment are used (Fig. 2): Hoyos Del Espino, Barco De Ávila, Puente Congosto, Salida Embalse de Santa Teresa,
Fresno-Alhandiga, Encinas de Arriba, Alconada, Salamanca and Contiensa. The information of discharges was extracted from

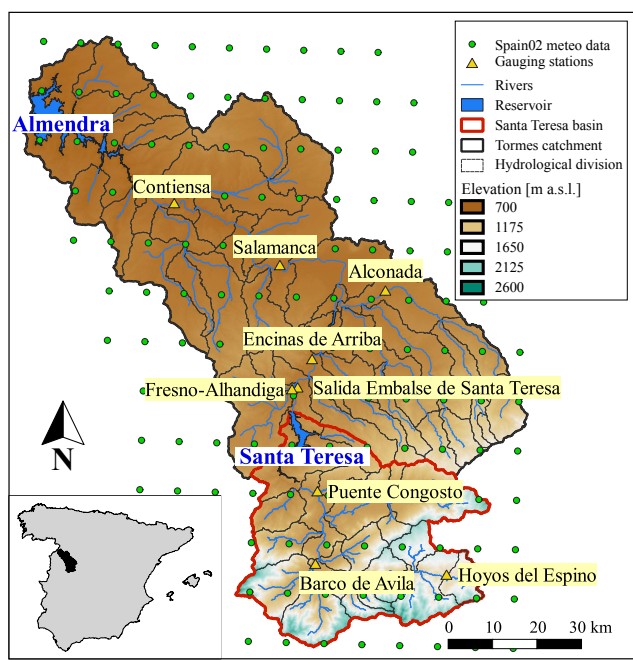

**Figure 2.** Location of the Santa Teresa and Tormes catchments, hydrological subdivision of the basin, reservoirs, gauging stations and the Spain02 gridded meteorological dataset.

the Center for Research and Experimentation of Public Works (CEDEX) platform (CEDEX, 2019). Moreover, a record of the historical water levels at the Santa Teresa reservoir from 1958 to 2015 is also available in this same platform.

## 4.2 Climate projections

The World Climate Research Programme (WCRP) Coordinated Regional Downscaling Experiment (CORDEX) project provides high-resolution regional climate projections and presents an interface for users of climate simulations in climate change impact, adaptation, and mitigation studies (Giorgi et al., 2009). As part of the CORDEX framework, the EURO-CORDEX initiative provides regional climate projections for Europe at 0.11° resolution (about 12 km) up to the year 2100 (Jacob et al., 2014). The regional simulations result from the downscaling of the Coupled Model Intercomparison Project Phase 5 (CMIP5) global climate projections (Taylor et al., 2012) and the Representative Concentration Pathways (RCPs) (IPCC, 2013; Moss et al., 2010).

In the present study, projections from the EURO-CORDEX project are used. These daily projections are available at the Earth System Grid Federation (ESGF) archiving system and accessible through one of its index nodes (e.g., ESGF Node, IPSL, 2019). In order to cover a large bandwidth of future climate evolutions, three different RCPs have been considered:

- RCP2.6: peak in radiative forcing at $\sim 3$ W/m$^2$ before 2100 and decline (van Vuuren et al., 2007, 2011).

- RCP4.5: stabilization without overshoot pathway to 4.5 W/m$^2$ at stabilization after 2100 (Thomson et al., 2011).

**Table 1.** List of climatic projections (CP) used in the study, indicating the driving GCM, ensemble member, institute and RCM for each of them, and which scenario is available.

| ID | Domain | Driving GCM | Ensemble | Institute | RCM | Historical | RCP2.6 | RCP4.5 | RCP8.5 |
|---|---|---|---|---|---|---|---|---|---|
| CP1 | EUR-11 | CNRM-CERFACS-CNRM-CM5 | r1i1p1 | CLMcom | CCLM4-8-17 | x | | x | x |
| CP2 | EUR-11 | CNRM-CERFACS-CNRM-CM5 | r1i1p1 | SMHI | RCA4 | x | | x | x |
| CP3 | EUR-11 | ICHEC-EC-EARTH | r12i1p1 | CLMcom | CCLM4-8-17 | x | x | x | x |
| CP4 | EUR-11 | ICHEC-EC-EARTH | r12i1p1 | KNMI | RACMO22E | x | x | x | x |
| CP5 | EUR-11 | ICHEC-EC-EARTH | r12i1p1 | SMHI | RCA4 | x | x | x | x |
| CP6 | EUR-11 | ICHEC-EC-EARTH | r1i1p1 | KNMI | RACMO22E | x | | x | x |
| CP7 | EUR-11 | ICHEC-EC-EARTH | r3i1p1 | DMI | HIRHAM5 | x | x | x | x |
| CP8 | EUR-11 | IPSL-IPSL-CM5A-LR | r1i1p1 | GERICS | REMO2015 | x | x | | |
| CP9 | EUR-11 | IPSL-IPSL-CM5A-MR | r1i1p1 | IPSL-INERIS | WRF331F | x | | x | x |
| CP10 | EUR-11 | IPSL-IPSL-CM5A-MR | r1i1p1 | SMHI | RCA4 | x | | x | x |
| CP11 | EUR-11 | MOHC-HadGEM2-ES | r1i1p1 | CLMcom | CCLM4-8-17 | x | | x | x |
| CP12 | EUR-11 | MOHC-HadGEM2-ES | r1i1p1 | DMI | HIRHAM5 | x | | | x |
| CP13 | EUR-11 | MOHC-HadGEM2-ES | r1i1p1 | KNMI | RACMO22E | x | x | x | x |
| CP14 | EUR-11 | MOHC-HadGEM2-ES | r1i1p1 | SMHI | RCA4 | x | x | x | x |
| CP15 | EUR-11 | MPI-M-MPI-ESM-LR | r1i1p1 | CLMcom | CCLM4-8-17 | x | | x | x |
| CP16 | EUR-11 | MPI-M-MPI-ESM-LR | r1i1p1 | MPI-CSC | REMO2009 | x | x | x | x |
| CP17 | EUR-11 | MPI-M-MPI-ESM-LR | r1i1p1 | SMHI | RCA4 | x | x | x | x |
| CP18 | EUR-11 | MPI-M-MPI-ESM-LR | r2i1p1 | MPI-CSC | REMO2009 | x | x | x | x |
| CP19 | EUR-11 | NCC-NorESM1-M | r1i1p1 | DMI | HIRHAM5 | x | | x | x |
| CP20 | EUR-11 | NCC-NorESM1-M | r1i1p1 | SMHI | RCA4 | x | | | x |
| CP21 | EUR-11 | NOAA-GFDL-GFDL-ESM2G | r1i1p1 | GERICS | REMO2015 | x | x | | |

– RCP8.5: rising radiative forcing pathway leading to 8.5 W/m$^2$ in 2100 (Riahi et al., 2007, 2011).

Moreover, the uncertainties inherent to the modelled temporal evolution of future climate will be tackled by using ensemble simulations that combine different RCMs with different GCMs, as it is done within the CORDEX framework.

Each projection also has a reference period or *Historical* simulation (1970-2005) needed to evaluate and eventually correct results based on the comparison against observed climatological data sets. Table 1 summarizes the 21 climate projections used in this study, indicating the driving GCM, the ensemble member, the institute that conducted the projection and the RCM for each of them, as well as the scenarios (Historical and RCP) available.

## 4.3 Dam risk model

As part of a quantitative risk analysis performed on 27 dams located in Spain (Ardiles et al., 2011; Morales-Torres et al., 2016), the individual risk model of the Santa Teresa dam was set up with iPresas software (iPresas, 2019) for hydrological loading scenarios. Such modelling was performed using event trees, an exhaustive representation of all the possible chains of events

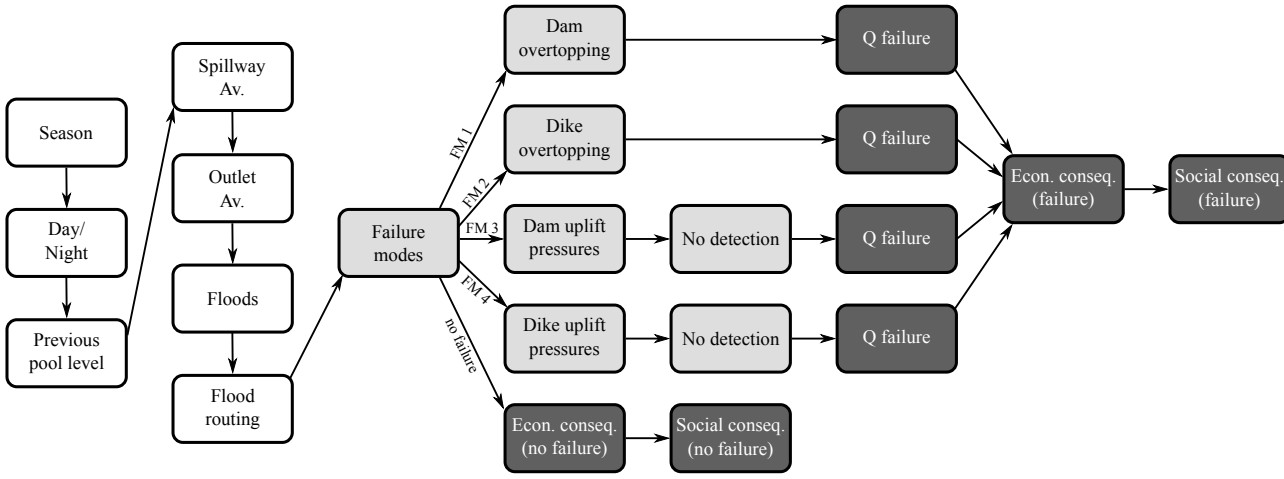

**Figure 3.** Diagram of the quantitative risk model for the Santa Teresa dam.

represented by nodes that can produce the dam failure (Serrano-Lombillo et al., 2011). The tree's branches represent all the possible outcomes of their event of origin and have an assigned probability. Any path between the initiating node and each of the nodes of the tree represent one of the possible outcomes that might result from the original event and can be calculated as the product of the probabilities associated with each branch (Fluixá-Sanmartín et al., 2019).

This model can be represented using the influence diagram presented in Fig. 3. As suggested in Fluixá-Sanmartín et al. (2018), this risk modelling approach is used in this work to structure and organize the assessment of the potential impacts of climate change on the different components of risk.

In the first five nodes the model defines the probability of different dam-reservoir system scenarios prior to the arrival of the largest flood of the year. This encompasses the probability of falling in a specific period of the year (*Season*), whether its day or night time (*Day/Night*), the annual exceedance probability curve of the water pool level of the reservoir (*Previous pool level*) and the probability of the bottom outlet works and spillways gates functioning properly (or not) when a flood arrives (*Spillway Av.* and *Outlet Av.*). The next node (*Floods*) introduces the flood entering the reservoir; a probabilistic hydrologic analysis is necessary to obtain the annual exceedance probability of potential incoming floods. The following node (*Flood routing*) includes the maximum pool levels and peak outflows resulting from the flood routing for each possible combination of previous pool level, inflow flood and gate availability.

The node *Failure modes* contemplates the four possible ways in which the Santa Teresa dam is supposed to fail: due to the overtopping of the dam or of the dike, or due to the sliding of the dam or of the dike. For each branch the model relates the maximum water level reached in the reservoir in each flood event with the conditional failure probability. It is worth noting that the sliding failure mode is decomposed in two nodes: the probability of being in different uplift pressures hypothesis (*Dam/Dike uplift pressures*) and the existing capacity to detect and to avoid high uplift pressures (*No detection*).

Finally, the following nodes are used to compute consequences in order to estimate risk, following Eq. (1). The nodes *Q fail* characterize the failure hydrograph for each failure mode by introducing a relation between the water pool level and the peak failure discharge. This relation was previously computed using hydraulic models of the dam breach.

Last nodes introduce the relation between the outflow hydrographs and the economic (*Econ. conseq. (failure)*) and loss of life consequences (*Social conseq. (failure)*). A common practice in dam safety is working with incremental consequences,
obtained by subtracting the consequences for the non-failure case to the consequences for the failure case (Serrano-Lombillo et al., 2011; SPANCOLD, 2012; USACE, 2011) in order to consider only the part of the incremental risk produced by the dam failure. Therefore, the consequences of the non-failure case (*Econ. conseq. (no failure)* and *Social conseq. (failure)*) must also be calculated to obtain incremental consequences.

## 4.4  Water resources management model

Risk modelling requires the analysis of the probability of occurrence of a certain water level in the reservoir at the moment of arrival of the flood. It defines the starting situation in the reservoir when studying the loads induced by the flood (SPANCOLD, 2012). Such analysis can be usually done by using the register of historic pool levels. However, the effects of climate change are expected to affect the future water availability mainly due to increased precipitation variability and potential evapotranspiration (IPCC, 2014). Therefore, the simulation of the system of water resources management under future conditions is necessary to
obtain the relation between water pool level and probability of exceedance.

The simulation consists of a sequential calculation of the allocation and use of the water resources based on the reservoir's exploitation rules. Apart from the evaluation of the future inflows of the system, this analysis requires as inputs the basin management strategy and the water demand that depends on the reservoir's supply. Such information is contained in the Hydrological Plan of the Duero River Basin (Confederación Hidrográfica del Duero, 2015) that describes the exploitation rules
of the 13 systems of the basin.

In particular, the Tormes system is composed of the Santa Teresa and the Almendra reservoirs of 496 hm$^3$ and 2649 hm$^3$ of volume capacity, respectively. The above-mentioned Hydrological Plan describes the water demands according to their category: agricultural (7), fish farming (5), urban (1) and industrial demands (1). The different demands of the Tormes system are mainly satisfied using the Santa Teresa reservoir according to the assignation rules established. It also specifies the minimum
ecological discharges at different points of the river that must be guaranteed through reservoir's releases. Figure 4 shows a schematic diagram with the distribution to each water demand and its return to the system according to the Hydrological Plan.

Another aspect considered is the limitation of water storage in the Santa Teresa reservoir. Given the seasonality of high flows entering the reservoir, the Hydrological Plan considers freeboard volumes that vary throughout the year to adapt to the expected incoming floods. The minimum and maximum volumes and their corresponding water levels to be ensured each
month in normal exploitation conditions are detailed in Table 2. These limitations are important for estimating water pool levels (Sect. 5.3.1). For this study, five periods of the year have been established from these specifications, coded as follows: *Dec-Feb* (December, January and February), *Mar* (March), *Apr* (April), *May-Nov* (May, June, October and November), and *Summer* (July, August and September).

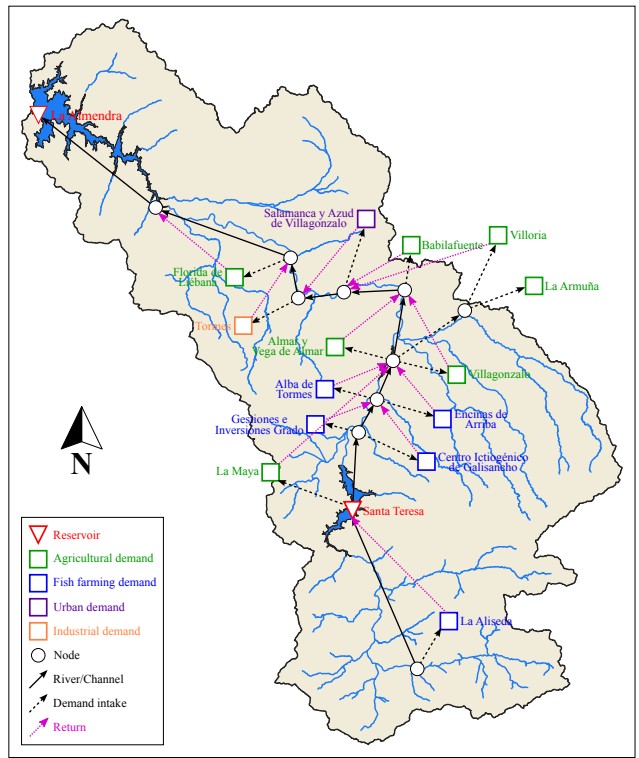

**Figure 4.** Scheme of the main elements of the Tormes Water Resources System.

## 5 Application of the methodology to the Santa Teresa dam

### 5.1 Correction of the RCM projections

Each precipitation and temperature projection described in Sect. 4.2 has been bias-corrected using a statistical transformation. In particular, an empirical non-parametric quantile mapping (eQM) approach (Boé et al., 2007; Panofsky and Brier, 1968) has been applied in this study using the R Software (R Development Core Team, 2008). This method has been widely applied in climatology and more detailed information can be found in the extensive literature (Cannon et al., 2015; Gudmundsson et al., 2012; Gutjahr and Heinemann, 2013; Maraun, 2016).

The goal is to define the transformation function for a modelled variable $x_{mod}$ so that its new distribution equals the distribution of the observed variable $x_{obs}$ corresponding to the reference period, as defined in Eq. (2):

$$x_{obs} = F_{obs}^{-1}(F_{mod}(x_{mod})) \tag{2}$$

where $F_{mod}$ is the empirical cumulative distribution functions (ECDF) of $x_{mod}$ and $F_{obs}^{-1}$ is the inverse ECDF (also named quantile function) corresponding to $x_{obs}$. In this case, the RCM-derived daily outputs represent the modelled variables while the daily data issued from the Spain02 v5 correspond to the observed variable.

**Table 2.** Seasonal minimum and maximum volumes (hm$^3$) and water levels (m a.s.l.) for the Santa Teresa reservoir.

| Month | Minimum volume (hm$^3$) | Maximum volume (hm$^3$) | Minimum level (m a.s.l.) | Maximum level (m a.s.l.) |
|---|---|---|---|---|
| January | 80 | 396 | 861.26 | 881.31 |
| February | 80 | 396 | 861.26 | 881.31 |
| March | 80 | 436 | 861.26 | 883.13 |
| April | 80 | 461 | 861.26 | 884.20 |
| May | 80 | 496 | 861.26 | 885.70 |
| June | 80 | 496 | 861.26 | 885.70 |
| July | 80 | 496 | 861.26 | 885.70 |
| August | 80 | 496 | 861.26 | 885.70 |
| September | 80 | 496 | 861.26 | 885.70 |
| October | 80 | 496 | 861.26 | 885.70 |
| November | 80 | 496 | 861.26 | 885.70 |
| December | 80 | 396 | 861.26 | 881.31 |

Once this transformation function has been defined, it is afterwards used to translate a simulated projection time series into a bias-corrected series. This procedure is applied separately for each climate projection (CP) described in Sect. 4.2 (Table 1) and for each of the three future *Periods* (1, 2 and 3), using the *Historical* period 1970-2005 as the calibration period of the correction function.

Corrected values in between fitted transformed values has been approximated using a linear interpolation. When model values from climate projections are larger than the training values used to estimate the ECDF, the correction found for the highest quantile of the training period is used (Boé et al., 2007; Jakob Themeßl et al., 2011).

In order to account for seasonally varying bias characteristics of the precipitation and temperature variables, the correction function itself has been determined separately for each season. Moreover, when correcting the precipitation projections, the number of wet days in the RCM time series of the *Historical* period has also been adjusted to fit the number of wet days in the observed time series of the same period.

Figure 5a shows an example of the empirical cumulative distribution functions (ECDF) corresponding to the Observed and the modelled CP3 Historical time series of daily temperature, for the grid cell with coordinates 40°05'60"N 5°48'00"W. The required shift towards the right (increase) of the CP3 series for an ECDF of 0.4 to match the observations has been highlighted with arrows. Figure 5b displays the bias-corrected temperatures (green line) from the original CP3 modelled time series (red line), compared to the observed series (blue line), for the year 1979.

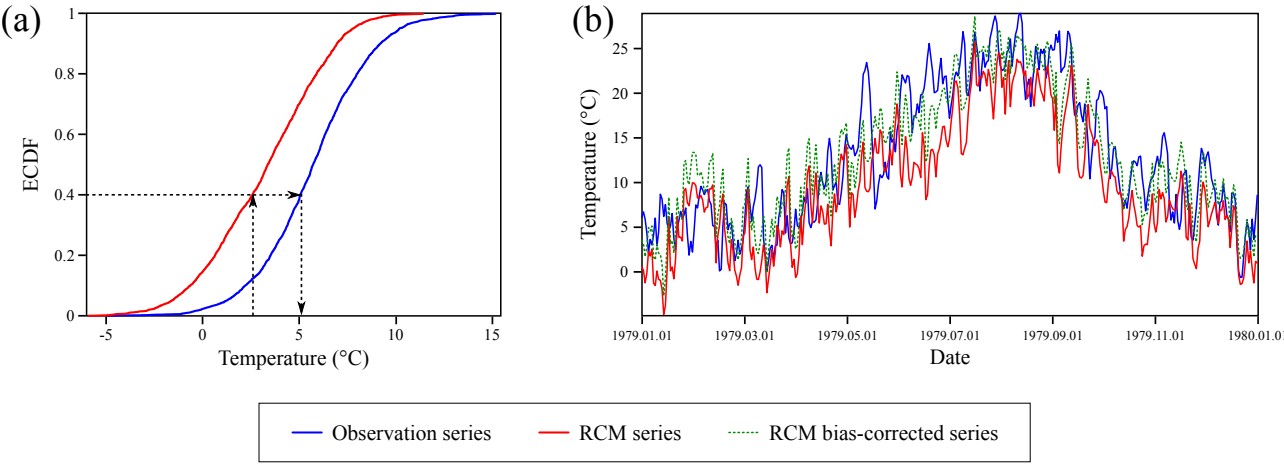

**Figure 5.** (a) Example of ECDF of the observed (blue) and the modelled CP3 (red) daily temperature series and bias correction using the eQM technique: the ECDF of the simulated series (red) is shifted to match with the observed ECDF (blue). (b) Time series of daily temperatures for the observed (blue) and the CP3 modelled (red) datasets, and bias-corrected series (green, doted line) for the year 1979.

## 5.2 Hydrological modelling

### 5.2.1 Setting and calibration of the model

A hydrological model of the Santa Teresa and the Tormes catchments has been elaborated with the hydrological-hydraulic modelling software RS MINERVE (Foehn et al., 2019; García Hernández et al., 2019), a freeware that allows rainfall-runoff calculations based on a semi-distributed concept and downstream propagation of discharges.

First, the basin has been divided in subbasins according to the hydrographic network and to the location of the gauging stations, as shown in Fig. 2. For this study, the GSM-SOCONT model (Schaefli et al., 2005) has been applied to each resulting

subbasin. Simulated natural processes use precipitation and temperature inputs to model surface and subsurface flow, infiltration, evapotranspiration, snow accumulation and melting. Channel routing of the rivers has been solved with the kinematic wave model, also available in RS MINERVE.

Finally, the model's calibration has been performed using the calibration module of the RS MINERVE software based on the observed records of the gauging stations described in Sect. 4.1. Calibrated parameters are: the reference degree-day snowmelt

coefficient (S); the maximum height of the infiltration reservoir (HGR3Max); the release coefficient of the infiltration reservoir (KGR3); and the runoff slope (J0) and Strickler coefficient (Kr) for the runoff surface as well as for the river reaches. The performance indicators used to assess the quality of the fit are Nash-Sutcliffe (Nash and Sutcliffe, 1970) and the Kling-Gupta efficiency (Gupta et al., 2009; Kling et al., 2012).

Periods with available discharge data are heterogeneous and thus calibration/validation processes have been adapted ac-

cordingly. It has been decided to use the period 01.10.2010-30.09.2015 as the calibration period, while the validation period depends on each gauging station. Results of the calibration/validation process for the gauging stations upstream of the Santa

**Table 3.** Calibration and validation results for the gauging stations upstream the Santa Teresa dam.

| Station | Calibration | | | Validation | | |
|---|---|---|---|---|---|---|
| | Period | Nash | Kling-Gupta | Period | Nash | Kling-Gupta |
| Hoyos del Espino | 01.10.2010-30.09.2015 | 0.612 | 0.749 | 01.10.1983-01.10.1995 | 0.570 | 0.726 |
| Barco de Ávila | 01.10.2010-30.09.2015 | 0.679 | 0.766 | 01.10.1971-30.09.1987 | 0.667 | 0.589 |
| Puente Congosto | 01.10.2010-30.09.2015 | 0.939 | 0.768 | 01.10.1997-30.09.2010 | 0.670 | 0.709 |

Teresa dam (Hoyos del Espino, Barco de Ávila and Puente Congosto) are presented in Table 3. Figure 6 shows the observed and modelled flows for these stations. For visualization purposes, only the period 01.10.2010-30.09.2015 is displayed. It is considered that the calibration presents adequate results for the purpose of the study.

### 5.2.2  Water management model simulation

The first purpose of the hydrological model of the Santa Teresa and Tormes catchments is the simulation of the water resources and its evolution with time. The basic inputs required are: (i) the reservoir's exploitation rules; (ii) the water demands; and (iii) the expected discharges at different points of the basin.

The first two inputs are extracted from the Hydrological Plan of the Duero River Basin (Confederación Hidrográfica del Duero, 2015) described in Sect. 4.3. For this study, the only demand that is considered variable with time is the urban demand, which corresponds to the supply to the city of Salamanca. This is a direct consequence of the population variation expected at this city which is further described in Sect. 5.3.4. For that, the individual consumption has been maintained and only the number of consumers has been adapted. In the absence of more detailed information, the rest of the demands (agricultural, industrial and fish farming) and the prioritization of the water supply for each demand (the importance and order in which each demand is satisfied) are assumed unaltered in future scenarios.

Concerning basin discharges, the hydrological model elaborated with RS MINERVE is able to simulate the rainfall-runoff processes at a daily resolution. Thus, the meteorological data issued from the Spain02 grid observations as well as from the climate projections are used as inputs to the model in order to obtain the consequent discharges at each subbasin (Fig. 2).

The simulation of the reservoir's response has been modelled including to the hydrological model different hydraulic elements available in RS MINERVE. On one hand, the water consumption has been modelled with *Consumer* objects which allow to define the flow abstraction series of each water demand (including the minimum ecological discharges) at each timestep. The order of preference defined in the Hydrological Plan guidelines for the supply to each demand has been respected. On the other hand, the outflows from the reservoir are managed using *Planner* objects: these models permit to create different rules that rest on the hydrological and hydraulic conditions of the basin. That is, the supply to a specific point depends on the demand at this point, the water level at the reservoir or the satisfaction of preferential demands. At this point, it is worth mentioning that the seasonal minimum and maximum levels contemplated by the Hydrological Plan (Table 2) have been incorporated to the model

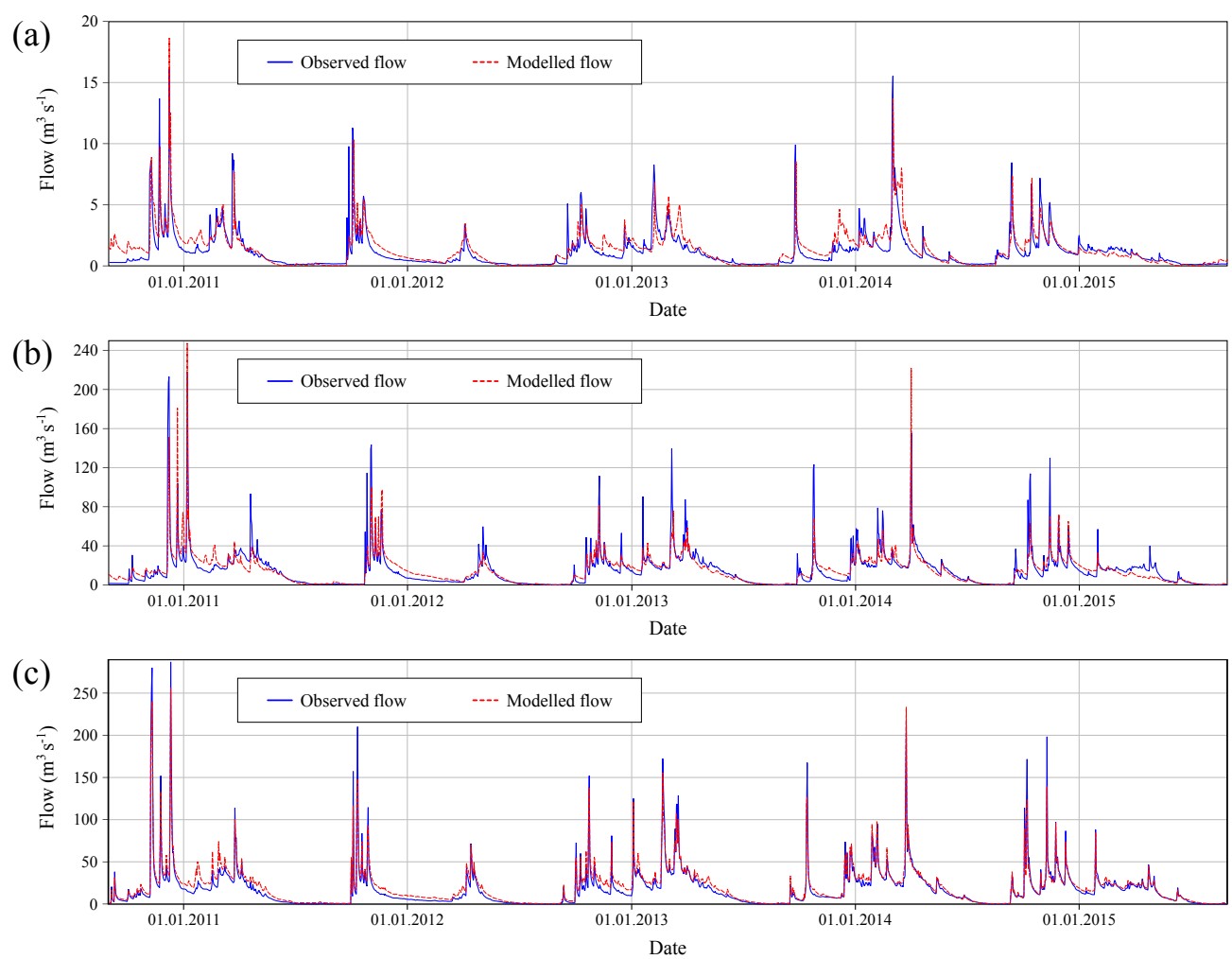

**Figure 6.** Comparison between observed (blue) and modelled (red) flows for the gauging stations (a) Hoyos del Espino, (b) Barco de Ávila, and (c) Puente Congosto.

within these *Planner* objects. More detailed descriptions on the use of such models can be found in (García Hernández et al., 2019).

The validation of this water resources model is conducted by comparing its results with a reference record. Figure 7 displays
285    the observed water levels recorded at the Santa Teresa reservoir and the simulated series obtained with the RS MINERVE model, for the period 1990-2015. As shown in the figure, results performance is moderate at the beginning of the period (1990-2000) and then increases notably from 2000 to 2015. This is mainly due to the fact that the reservoir's exploitation rules used in the model are based on the last Hydrological Plan of the basin (Confederación Hidrográfica del Duero, 2015), which is relatively recent. It is likely that before 2000 the operational rules were different and thus the model is not capable of capturing
290    the real fluctuations of the water resources. For the purposes of the study, it is considered that the overall performance of the

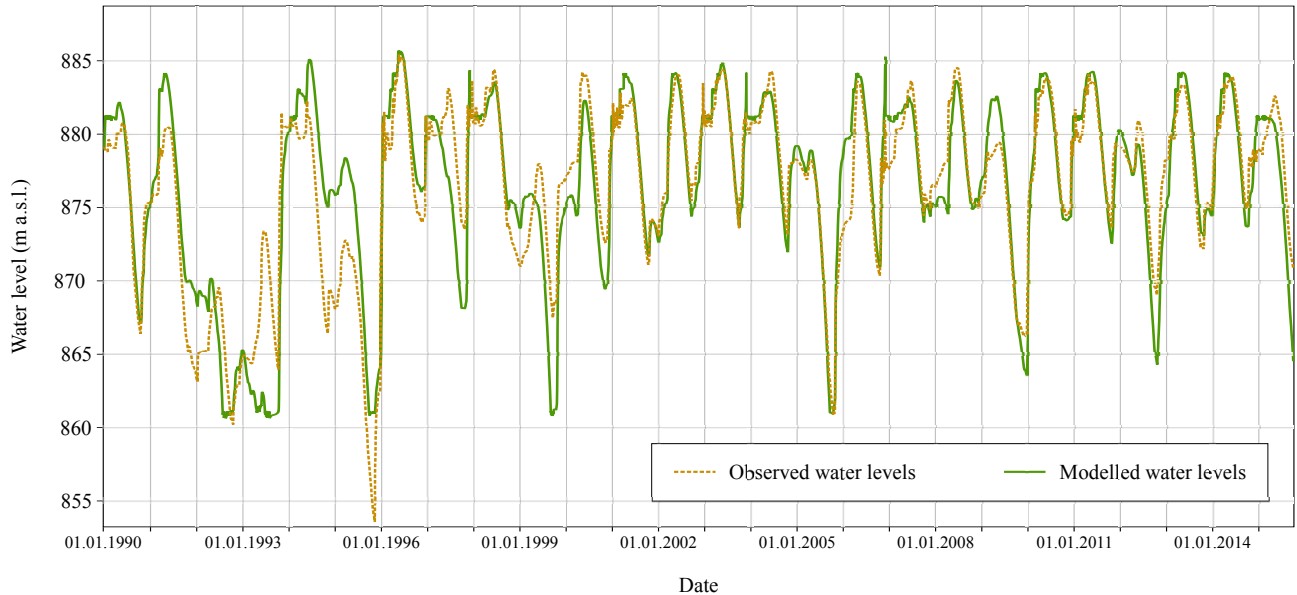

**Figure 7.** Observed and simulated water levels at the Santa Teresa reservoir, between 1990 and 2015.

hydrological model is adequate to simulate the water resources at the Santa Teresa reservoir. Once the model is validated, the different simulations have been processed for the *Historical* and the future periods.

### 5.2.3 Design flood hydrographs

Additionally, the hydrological model has been employed for the definition of the design flood hydrographs entering the Santa Teresa reservoir. A deterministic approach based on the design storm method (ASCE, 1996; Reed et al., 1999) has been followed. In this method, a design storm is defined based on the intensity duration frequency (IDF) curve of rainfall and applied to an event-based hydrological model to calculate the hydrographs. Statistical methods have been discarded mainly due to a lack of representative flood records, in particular for the characterisation of future floods. The process consists of three main parts: the generation of synthetic storms, the definition of the initial conditions of the basin, and the simulation of the flood hydrographs. What follows is a detailed description of these steps. It is worth mentioning that the process has been individually applied to the different periods considered (*Historical*, *1*, *2* and *3*) in order to assess the changes in the resulting floods from the Base Case until the end of the 21$^{st}$ century.

Generation of design storm hyetographs

The definition of the design storm hyetograph first requires the statistical analysis of the annual maxima of storm rainfall, extracted from the daily precipitation data of the observation and climate projection series for each point of the Spain02 grid. This allows to obtain the maximum daily precipitation for any return period considered. Each annual maxima series has been fitted to a Gumbel distribution, a widely used option in the Spanish territory. Once the distribution fitted, the daily precipitations corresponding to the following return periods have been calculated: 2, 5, 10, 25, 50, 100, 200, 500, 1 000, 2 000, 5 000, 10

000, 20 000, 50 000 and 100 000 years. In order to evaluate the sensitivity of risk results to the fitted Gumbel distribution, a complementary sensitivity analysis is included in Appendix A

Then, a predefined IDF curve has been used to estimate the rainfall depth for any given duration and for the selected return periods. The formulation of the IDF curve is taken from the document of Ministerio de Fomento (2016) and is expressed as:

$$\frac{I_t}{I_d} = \left(\frac{I_1}{I_d}\right)^{\frac{28^{0.1}-t^{0.1}}{28^{0.1}-1}} \tag{3}$$

Where $I_t$ is the average intensity (in mm/h) corresponding to the time interval of duration $t$; $I_d$ is the daily average intensity (in mm/h) corresponding to the return period considered, and equal to $P_d/24$; $P_d$ is the total daily precipitation (in mm) corresponding to the return period considered; $I_1/I_d$ is the ratio between the hourly and daily intensity, obtained from Ministerio de Fomento (2016) and equal to 10.2 for the study case.

It is worth mentioning that this formula is climate- and location-dependent since it has been extracted from an analysis based on historical records. However, in this study the formula has also been applied for future climatic conditions. The difficulty to establish IDF relations with no sub-daily precipitation data available is one of the limitations of the present work. Thus, in order to deal with this issue, the option chosen was to rely on pre-defined formulations such as the one presented in Eq. (3).

Temporal rainfall distribution is obtained using the alternating block method (Chow et al., 2008), where the intensity of each time interval is read from the previous IDF curve. Subsequently, the rainfall depths for each interval (P1, P2, ...) are obtained taking the difference between successive rainfall depth values, with $\Delta t = 0.5$ h. The blocks P1, P2, ... are reordered with the maximum intensity at the centre of the hyetograph and the other blocks alternating to the right and left. In the absence of more detailed, it has been considered that the duration of the storm events is 24 hours.

Given that rainfall is never evenly distributed over the area of study due to the topographic variability of the catchment areas, the use of an Areal Reduction Factor (ARF) is required to correct each grid point rainfall and avoid an overestimation of the rainfall input. The ARF adopted follows the empirical formulation proposed in (Témez, 1991) for the Spanish territory:

$$ARF = 1 - \frac{log(A)}{15} \tag{4}$$

where $A$ is the area of the catchment (in km$^2$). In this case, the drainage area of the Santa Teresa reservoir is 1853 km$^2$.

Initial basin conditions

Francés et al. (2012) and Rogger et al. (2012) highlighted an important drawback when applying the design storm method. It is generally assumed that the rainfall and the discharge return periods are equal, and no other factors such as the initial conditions of the basin are generally considered. Indeed, the proper selection of basin antecedent conditions is of paramount importance for the runoff definition.

To address such limitation, an analysis of three different state variables of the hydrological model was performed: the level in the infiltration reservoir (HGR3), the runoff water level downstream of the surface (Hr) and the river discharge (Q). The goal was to define a characteristic initial state of the basin prior to the occurrence of each storm.

Once the hydrological model set, it was used to run the rainfall-runoff simulations corresponding to the different scenarios (observations and projections) and for all the periods considered (*Historical*, *Period 1*, *Period 2* and *Period 3*). For each

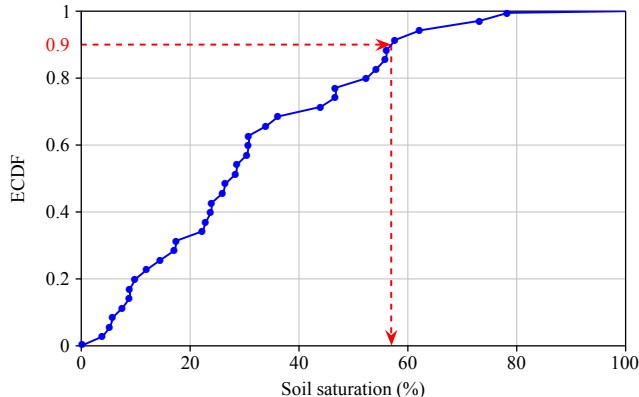

**Figure 8.** Example of ECDF curve for the soil saturation (relative HGR3 state variable) and extraction of the value corresponding to a non-exceedance probability of 0.9.

simulation, the dates on which the annual maximum rainfalls occurred were identified, allowing to extract the state variables of the model corresponding to the precedent day. This resulted in a set of state variables per year for each simulation. From each of these series of state variables, an ECDF curve was generated. In this way, the initial conditions matching with the storm

345 hyetograph of return period T can be obtained reading from the ECDF curve the value for a non-exceedance probability equal to *1-1/T*. Figure 8 illustrates the extraction of the soil saturation (calculated as *HGR3/HGR3Max×100* for the SOCONT model) corresponding to a non-exceedance probability of 0.9 or a return period of 10 years.

Hydrograph calculation

The model developed with RS MINERVE and described above was used as the event-based hydrological model to simulate

350 the behaviour of the Santa Teresa basin. In this case, the simulation timestep was set at 10 minutes in order to better capture the hydrological processes occurring in the basin. Once each storm hyetograph and set of initial conditions corresponding to a return period between 2 and 100 000 years has been defined, the model was run, and the flood hydrographs are obtained.

Resulting floods for the Base Case are presented in Fig. 9a. Peak discharge by return period is displayed in Fig. 9b.

## 5.3   Risk modelling

355 Considering the exposure of the dam to climate change, the risk model of the Santa Teresa dam (Fig. 3) is updated following the effects of climate change on each of the risk components. Among these components, mainly four have been identified as susceptible to be altered: previous pool levels in the reservoir, spillway gate and bottom outlet performance, floods entering the reservoir and social consequences used to compute the social risk. In the absence of more detailed analyses, in this study other risk model components are assumed unaltered.

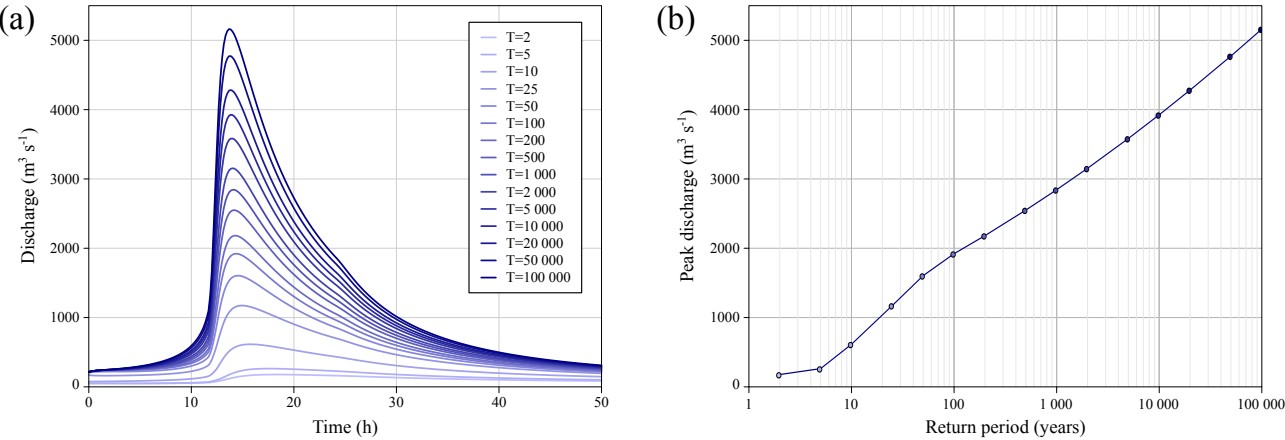

**Figure 9.** (a) Resulting flood hydrographs for return periods between 2 and 100 000 years, for the Base Case. (b) Flood frequency characterization of the maximum values of peak discharges.

### 5.3.1 Previous pool level

Based on the reservoir levels obtained from the water resources simulation of each scenario defined in Sect. 5.2.2, the empiric exceedance probability curve of the pool levels is obtained by ordering all the data in an increasing order (SPANCOLD, 2012) and applying Eq. (5):

$$PE_n = 1 - \frac{i_n - 1}{N - 1} \tag{5}$$

where $PE_n$ is the probability of exceedance for a pool level $n$, $i_n$ is the number of order of pool level $n$ within the series of sorted levels and $N$ is the length of the series.

The resulting curve is discretized in different not equidistant intervals to be included within the risk model event tree. In the event tree, the probability of each branch is the probability of falling within any of the values of the interval considering a representative value of each interval – usually the average value of the interval – . Since the risk model used in this study considers the specific period of the year in which the flood occurs, the reservoir's exploitation rules differ depending on this period (Sect. 4.3) and thus imply different exceedance probability curves of the pool levels. The analysis of the previous pool level must therefore be done for each of the periods considered. Figure 10 shows the comparison of the exceedance probability curves corresponding to the Base Case and to the climate scenario CP1 (RCP45 and *Period 1*), both computed for the *Summer* season. As can be appreciate, the results of the CP1 projection present lower water levels than for the Base Case. This is mainly due to the reduction in the discharge contributions to the reservoir and the enhanced evapotranspiration directly related to the increase of temperatures.

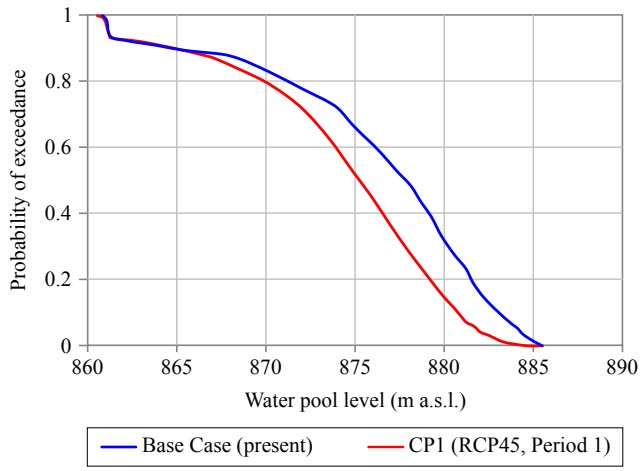

**Figure 10.** Relation between water pool level and probability of exceedance for the Base Case (present situation) and the climate projection CP1 (RCP45 and *Period 1*), for the *Summer* season.

### 5.3.2 Gate performance

In the context of dam safety, spillways and outlet works play a fundamental role. The estimation of their reliability, i.e., that in the moment of the arrival of the flood they can be used, makes part of the studies required to feed a risk model. In a basic analysis, individual reliability can be estimated directly for each gate using the qualitative description of the gate system's condition. Escuder-Bueno and González-Pérez (2014) propose a classification based on these descriptors that avoids resorting to detailed studies such as fault trees:

- 95%: the outlet is new or has been very well maintained.

- 85%: the outlet is well maintained but has had some minor problems.

- 75%: the outlet has some problems.

- 50%: the outlet is unreliable for flood routing.

- 0%: the outlet is not reliable at all or it has never been used.

In this analysis, gates can be considered independent and thus the probability of each availability gates case can be estimated with a binomial distribution (Eq. (6)):

$$P(x) = \frac{n!}{x! \cdot (n-x)!} \cdot p^x \cdot (1-p)^{n-x} \tag{6}$$

Where *P(x)* is the probability that *x* number of gates work properly, *n* is the total number of gates and *p* is the individual reliability of gates.

As part of the quantitative risk analysis performed on the Santa Teresa dam, the state of the spillway gates and the bottom outlet was estimated as well maintained. Their individual reliabilities were thus established as 85% for the present situation. However, the conditions of the gates can deteriorate with time and with changing hydro-meteorological conditions. As mentioned in Fluixá-Sanmartín et al. (2018), certain factors as increased soil erosion due to more intense rainfalls or greater fluctuations in temperature could eventually lead to a decreased reliability of the gates. In this study, the state of both the spillway and the bottom outlet gates is assumed to progressively deteriorate until the end of the 21$^{st}$ century. Following a simple approach, it is considered that some problems may appear and thus the individual reliability will vary from 85% to 75%, corresponding to *Period 3* (2070-2099). For the intermediate scenarios a linear interpolation is applied to obtain the individual reliability, that is 81.5% for *Period 1* (2010-2039) and 78.5% for *Period 2* (2040-2069).

### 5.3.3 Floods

Since the present study analyses the risk of the dam under a hydrological scenario, it is supposed that the floods are the main loads to which the dam is subjected. Therefore, the resulting flood hydrographs obtained in Sect. 5.2.3 have been incorporated to update the risk model of the dam. As described above, each hydrograph is characterized by its return period or annual exceedance probability which defines the probability associated to each branch of the risk model emerging from the *Floods* node (Fig. 3). This also has an impact on the outcomes of the dam's flood routing, in particular the maximum pool levels and the peak outflows. It has been considered however that the flood routing strategy remains unchanged as defined in the Operation Rules document of the dam.

### 5.3.4 Social consequences

The dam risk model used in this study considers the social consequences resulting from the dam failure (Fig. 3) which rely on the exposure of people in the at-risk area to the dam output hydrograph. These consequences correspond to the number of fatalities among the inhabitants of the different population nucleus between the Santa Teresa and the Almendra dams.

Under future scenarios, the evolution of population at risk is thus expected to affect the potential casualties and needs to be considered to adequately assess the social risk. This does not account for a direct effect of climate change; however, this non-climatic factor has been considered in this study in order to contemplate a more realistic situation in future scenarios.

For this analysis, the long-term population projections at national scale available in the online publication Our World in Data (2018) extracted from the UN database (United Nations, 2017) were used. According to these projections, population is expected to slightly decrease until 2040 and will follow a substantial diminution until the end of the century. It has been supposed that the same pattern at the national level can be replicated at the regional and local levels. Therefore, in order to adapt the dam risk model used, the population at risk at the different cities and settlements has been proportionally reduced under the three future scenarios envisaged. Hence, the relative variation compared to the population in 2010 is as follows: -2.52% for *Period 1* (2010-2039); -14.37% for *Period 2* (2040-2069); and -22.25% for *Period 3* (2070-2099).

It is worth mentioning that, for the assessment of the economic consequences, the same current assets and services at risk remain so in the future and no new services are considered. Moreover, their economic cost has not been updated in order to work only with present values, independently of the future scenario considered.

## 6 Results and discussion

Once the dam risk model is adapted following the effects of climate change on each of the risk components, the social and economic risks [consequences/year] are calculated for the Base Case and for all the CP-period-RCP combinations. For the Base Case (present situation), the failure probability is $2.91 \times 10^{-6}$/year, while the social and economic risks are $2.56 \times 10^{-4}$ lives/year and $7.53 \times 10^{-4}$ M€/year respectively.

The evolution of social and economic risks for each RCP, from the present situation until the end of the 21st century, is presented in Fig. 11. For illustrative purposes, the y-axis is plotted on a logarithmic scale to better appreciate the order of magnitude of its values. The dashed black line indicates the present risk and helps highlighting whether the future risk of a particular CP is above or under such reference risk level. In general, these results indicate that in most future scenarios a deterioration of both the social and economic risks occurs. Indeed, the risk tends to increase in comparison to the present risk level and a certain dispersion of the risk appears with time. However, the RCP8.5 cases present a wider dispersion of results and no homogeneous effects can be extracted from it.

In order to deepen in the analysis, the resulting risks have been decomposed in its associated probability of failure and average consequences. Figure 12 represents this disaggregation of social and economic risks for each period considered. In such graph, risk is the dimension that combines both axes and is smaller in the lower left corner and grows towards the upper right corner. This is a widely used type of representation, used for instance by the US Bureau of Reclamation (USBR, 2011) to propose tolerability recommendations for incremental risk. Logarithmic scales are used in both axes and the same legend as in Fig. 11 is applied for the points. The present risk level has been represented as a black point and its probability of failure and consequences are highlighted with two dashed black lines. These lines divide the graph in four quarters labelled as:

- Type I: cases where the failure probability is greater, and the consequences are lower than in the Base Case.

- Type II: cases where both the failure probability and the consequences are greater than in the Base Case.

- Type III: cases where both the failure probability and the consequences are lower than in the Base Case.

- Type IV: cases where the failure probability is lower, and the consequences are greater than in the Base Case.

Moreover, Table 4 and Table 5 present the percent of cases falling in each of these situations, grouped by period and RCP. These results exhibit a tendency of the cases analysed to be in the Type I, and a lower proportion in the Type III situation, for all the periods analysed. Therefore, most cases indicate a reduction in the average consequences (not only due to the diminished exposure of people in the at-risk area) as well as an increase of the probability of failure of the dam.

**Table 4.** Percent of social risk cases falling in each Type (I, II, III or IV) grouped by period and RCP.

| Period | RCP | Type I | Type II | Type III | Type IV |
|--------|-----|--------|---------|----------|---------|
| | RCP2.6 | 55% | 0% | 45% | 0% |
| 2010-2039 | RCP4.5 | 82% | 0% | 6% | 12% |
| | RCP8.5 | 63% | 0% | 11% | 26% |
| | RCP2.6 | 91% | 0% | 9% | 0% |
| 2040-2069 | RCP4.5 | 88% | 0% | 12% | 0% |
| | RCP8.5 | 68% | 0% | 26% | 5% |
| | RCP2.6 | 91% | 0% | 9% | 0% |
| 2070-2099 | RCP4.5 | 76% | 0% | 24% | 0% |
| | RCP8.5 | 58% | 0% | 42% | 0% |

Since in this study the different components of the risk model have been adapted and analysed concurrently (Sect. 5.3), risk results do not highlight the individual contribution of each component to the final risk state. However, the use of risk models allows to decompose the contribution of each node in the final risk. For this purpose, a sensitivity analysis has been performed on the different risk components (*Previous pool level*, *Gate performance*, *Floods* and *Social consequences*) and their effect on the final dam failure risk, comparing to the overall effects combined. Results are presented in Fig. 13. According to these results, the *Floods* component has the larger influence on increasing the final risk. Furthermore, for its part the *Previous pool level* component tends in general to lower the risk in all cases. And as expected, deterioration in gate performance makes both risks to increase, mainly due to an increase in the failure probability. Therefore, the effects of climate change on the dam failure risk are mainly explained by the changes in the flood loads and the changes in the reservoir water levels regime. This explains the differences between each RCP scenario. Indeed, as the emission scenario worsens (from RCP2.6 to RCP8.5) the discharge contributions and especially the higher evapotranspiration related to the increase of temperatures are expected to reduce the water levels in the reservoir. This will ultimately cause a more marked worsening of the risk for the RCP2.6 scenario than for the RCP8.5 scenario.

Although a general increase of the risk can be extracted from the results, it is difficult to directly define unequivocal recommendations for dam owners and managers. Different factors play important roles when assessing risk management action plans: Are risk acceptable in present situation? Are they acceptable in future scenarios? What are the risk reduction measures envisaged? How long should we wait before implementing them? What is the efficiency of each of these measures? What criteria should we follow to prioritize them? In order to exploit these results in the context of decision-making support, further efforts to address the non-stationarity nature of risk as well as its intrinsic uncertainties are needed. Such issues impose a deeper evaluation of the recommendations to make for the development of long-term adaptation strategies. This line of research is in progress and still has the potential for improving comprehensive decision-making support based on future changes in dam risk.

**Table 5.** Percent of economic risk cases falling in each Type (I, II, III or IV) grouped by period and RCP.

| Period | RCP | Type I | Type II | Type III | Type IV |
|--------|------|--------|---------|----------|---------|
| | RCP2.6 | 55 % | 0 % | 36 % | 9 % |
| 2010-2039 | RCP4.5 | 76 % | 6 % | 0 % | 18 % |
| | RCP8.5 | 63 % | 0 % | 5 % | 32 % |
| | RCP2.6 | 82 % | 9 % | 9 % | 0 % |
| 2040-2069 | RCP4.5 | 88 % | 0 % | 0 % | 12 % |
| | RCP8.5 | 68 % | 0 % | 5 % | 26 % |
| | RCP2.6 | 82 % | 9 % | 0 % | 9 % |
| 2070-2099 | RCP4.5 | 76 % | 0 % | 12 % | 12 % |
| | RCP8.5 | 58 % | 0 % | 32 % | 11 % |

# 7 Conclusions

This article presents a comprehensive quantitative assessment of the effects of climate change on the failure risk of the Santa Teresa dam under hydrological scenarios, i.e. where the floods are the main loads to which the dam is subjected. The analysis integrates the various projected effects acting on each component of the risk, and how the dam failure risk evolves until the end of the 21$^{st}$ century.

The analysis is based on existing data and models from different sources. In particular, the climate projections (CPs) extracted from the CORDEX project have been treated and adapted for the study case. In order to deal with the associated uncertainty of climate modelling issued from the dispersion of their projection, the analysis is applied to the 21 available CPs. Additionally, a hydrometeorological model have been elaborated to simulate the response of the studied basin to present and future climatic conditions. Finally, the risk model of the dam has been adapted to the new components issued from the climate change impacts. Figure 1 summarizes the methodology proposed.

Results show a significant uncertainty of risk given by the dispersion of climate projection inputs and by the sensitivity to the hydrological modelling. In general, results show in most future scenarios an increase of both the social and economic risks in comparison to the present risk level, especially for the RCP2.6 and RCP4.5 scenarios. Moreover, most cases indicate a reduction in the average consequences as well as an increase of the probability of failure of the dam.

The use of a dam risk model allowed integrating the expected effects of climate change on the different components of the dam risk. The sensitivity analysis performed has shown that the effects of climate change on the dam failure risk are mainly explained by the changes in the flood loads and the changes in the reservoir water levels regime.

The methodology presented in this paper can serve as a useful guidance for dam owners and dam safety practitioners in the analysis of other study cases by entailing different models and data sources. This would eventually allow a more efficient planning of dam safety investments on the long term and even the adaptation of existing dam exploitation rules. New approaches that take into account the evolution with time of risk and of the efficiency of measures are thus needed. Furthermore, it is

important to highlight that, without the use of risk models, the integration of the various projected effects of climate change on each dam safety aspect would not have been possible.

In conclusion, the methodology proposed in this paper allows a detailed quantification of the effect of climate change on dam safety, which is one of the main concerns of managers and technicians of these critical infrastructures for water supply and energy production worldwide. However, in order to exploit such results in the context of decision-making support, further efforts to address the non-stationarity nature of risk as well as its intrinsic uncertainties are needed. Such issues impose a deeper evaluation of the recommendations to make for the development of long-term adaptation strategies.

*Data availability.* We thank AEMET and the University of Cantabria for the data provided for this work (Spain02 v5 dataset, available at AEMET (2019)). The hydrological information used in this study is available at the CEDEX platform (CEDEX, 2019).

## Appendix A:  Sensitivity analysis to precipitation Gumbel distribution

The use of precipitation data from the observation and climate projection series induces sampling errors in estimating the Gumbel probability distribution parameters applied in Sect. 5.2.3, which induces uncertainty to the estimated quantile-frequency relationship. This will eventually impact on the estimated dam failure risk, as a result of the methodology proposed above. In this Appendix, the influence of the Gumbel distribution fitting uncertainty on the estimated dam risks is investigated. A sensitivity analysis has been applied to the Base Case (present situation); this would give an idea on how the rest of cases would react under the same uncertainty.

It can be assumed that, due to the sampling error, the $T$ year quantile estimator $x_T$ of daily precipitation can be treated as a random variable (Su and Tung, 2013), as shown in Fig. A1. In this paper, the maximum likelihood method proposed by Kite (1988) is applied to calculate the sampling error of the Gumbel-based quantile estimator. According to such method, the variance for the Gumbel $T$ year quantile estimator ($x_T$) can be expressed as:

$$s_e^2(x_T) = \frac{\beta^2}{n} \cdot (1.1087 + 0.5140 \cdot Y + 0.6079 \cdot Y^2) \tag{A1}$$

Where $\beta$ is the scale parameter of the fitted Gumbel distribution, *n* is the sample size and *Y=-ln(-ln(1-1/T))*.

Then, assuming the sampling distribution of the *T* year quantile estimator to be normal (Kite, 1975; Su and Tung, 2013) with mean $x_T$ and variance $s_e^2$, 200 random quantiles are generated for the observation series (Base Case) for each return period $T$. Thenceforward, the corresponding hydrographs are obtained by replicating the process described in Sect. 5.2.3, this time using the new quantile-frequency relationships to determine the daily precipitations corresponding to return periods between 2 and 100 000 years. Finally, the risk model is applied, and social and economic risks are obtained for each of the 200 aleatory cases.

Results are displayed in Fig. A2. Risks have been decomposed in its associated probability of failure and average (a) social and (b) economic consequences. Moreover, the point densities for consequences (x-axis) and failure probability (y-axis) are obtained by applying the kernel density estimation technique (Parzen, 1962; Rosenblatt, 1956) and displayed in red. The Base

Case risk is represented as a black point and its probability of failure and consequences are highlighted with two dashed black lines.

Results show a significant sensitivity of risks to the meteorological modelling, and in particular to the statistical distribution fitting used to obtain maximum daily precipitations. Failure probability varies from $3\times10^{-7}$/year to $3\times10^{-5}$/year (being $2.91\times10^{-6}$/year the probability of the Base Case), that is two magnitude orders. Social and economic risks fluctuate between $5.64\times10^{-5}$ and $2.33\times10^{-3}$ lives/year, and between $1.06\times10^{-4}$ and $7.59\times10^{-3}$ M€/year, respectively. It is worth noting that the peak density for both the social and economic consequences is approximately coincident with the corresponding to the Base Case.

*Author contributions.* Javier Fluixá-Sanmartín prepared the paper with contributions from all co-authors.

*Competing interests.* The authors declare that they have no conflict of interest.

*Acknowledgements.* The authors acknowledge the Spanish Ministry for the Ecological Transition (MITECO) for its support in the preparation of this paper.

We acknowledge the World Climate Research Programme's Working Group on Regional Climate, and the Working Group on Coupled Modelling, former coordinating body of CORDEX and responsible panel for CMIP5. We also thank the climate modelling groups (listed in Table 1 of this paper) for producing and making available their model output. We also acknowledge the Earth System Grid Federation infrastructure an international effort led by the U.S. Department of Energy's Program for Climate Model Diagnosis and Intercomparison, the European Network for Earth System Modelling and other partners in the Global Organisation for Earth System Science Portals (GO-ESSP).

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

735

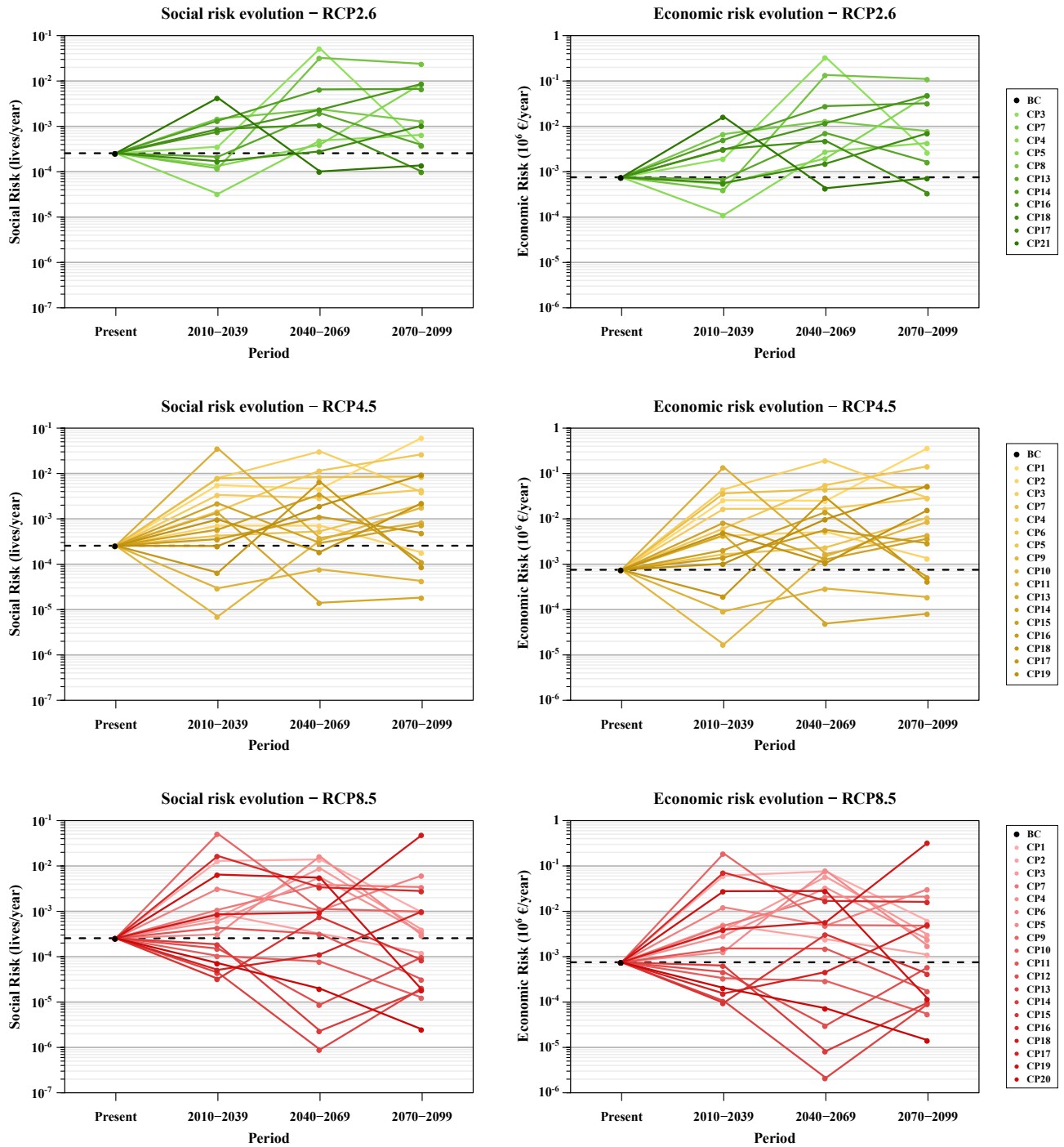

**Figure 11.** Social and economic risk results (left and right graphs, respectively), classified by RCP. The Base Case (BC) situation is highlighted with a black point and a dashed line.

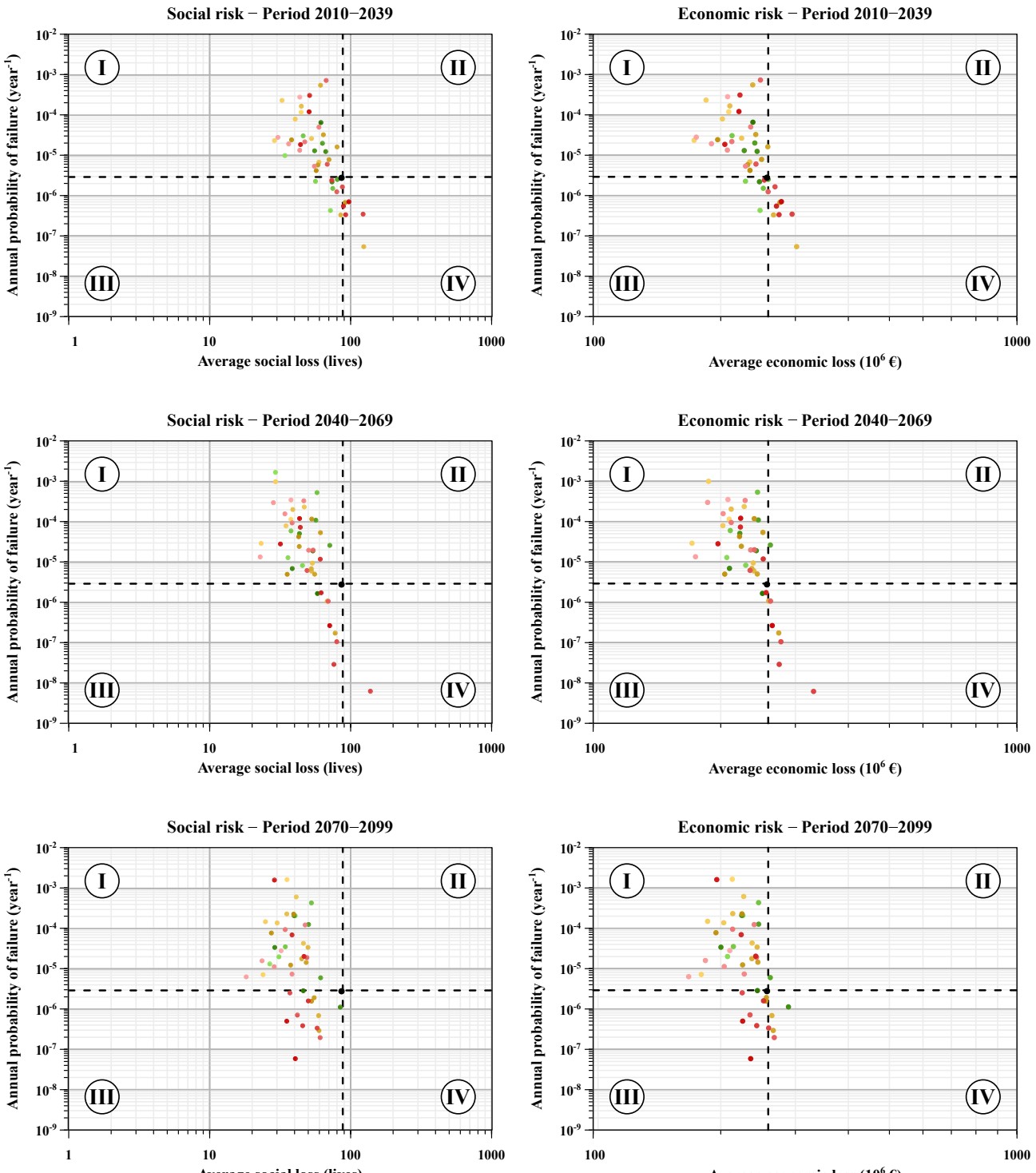

**Figure 12.** Disaggregation of social and economic risks (left and right graphs, respectively) in annual probability of failure and average consequences, classified by simulation period. The same legend as in Fig. 11 is applied here for the points. The Base Case situation is highlighted with a black point and two dashed lines.

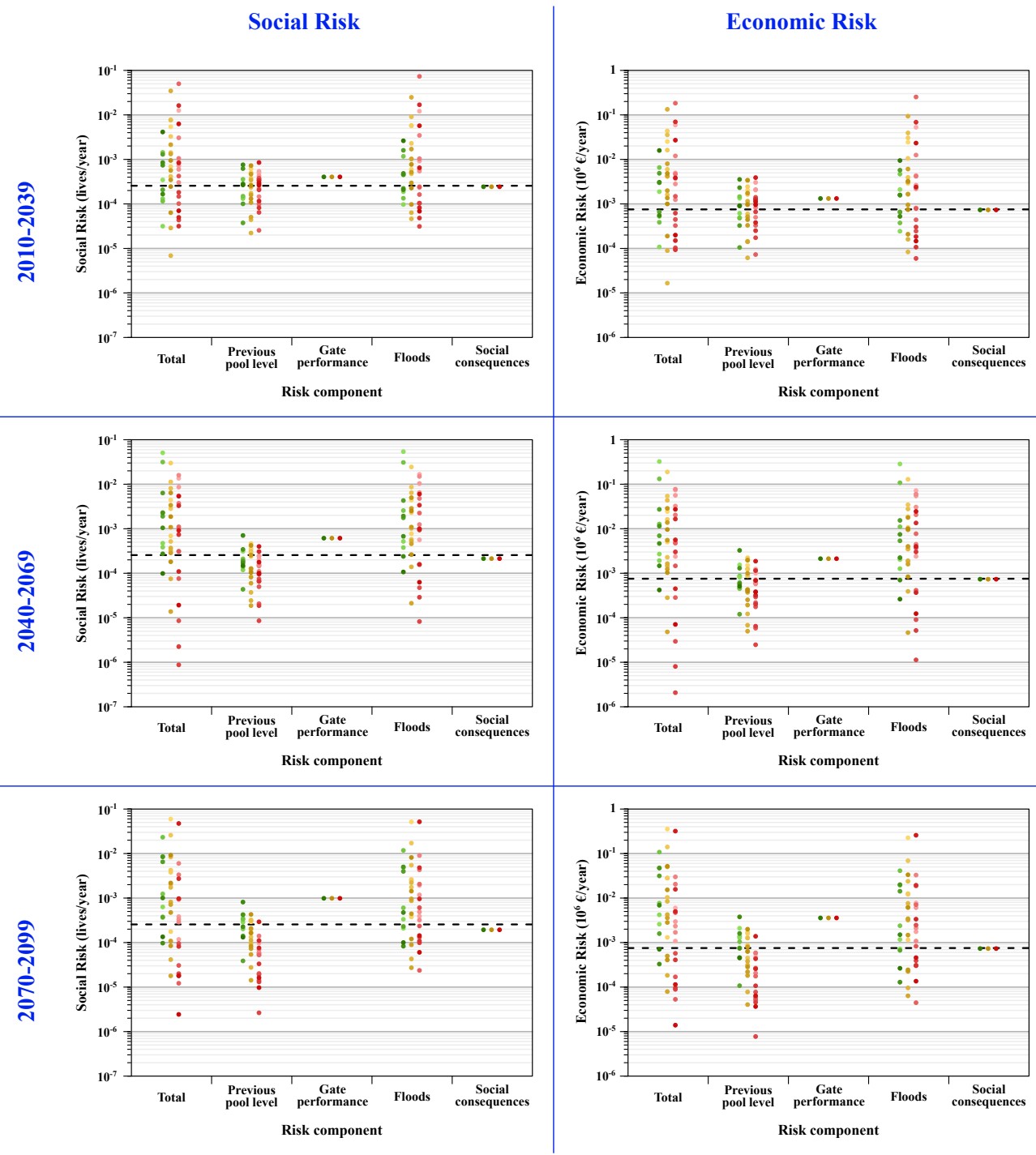

**Figure 13.** Individual effects of each risk model component on the total social and economic risk computed, classified by period. The same legend as in Fig. 11 is applied here for the points. The Base Case situation is highlighted with a black dashed line.

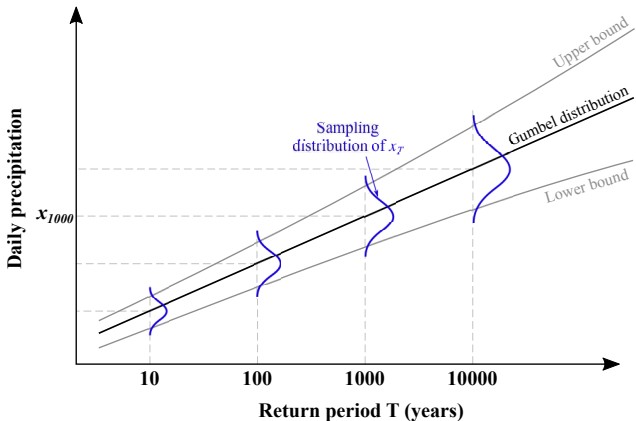

**Figure A1.** Uncertainty of estimated T year daily precipitation quantile due to sampling error (adapted from Kite (1975)).

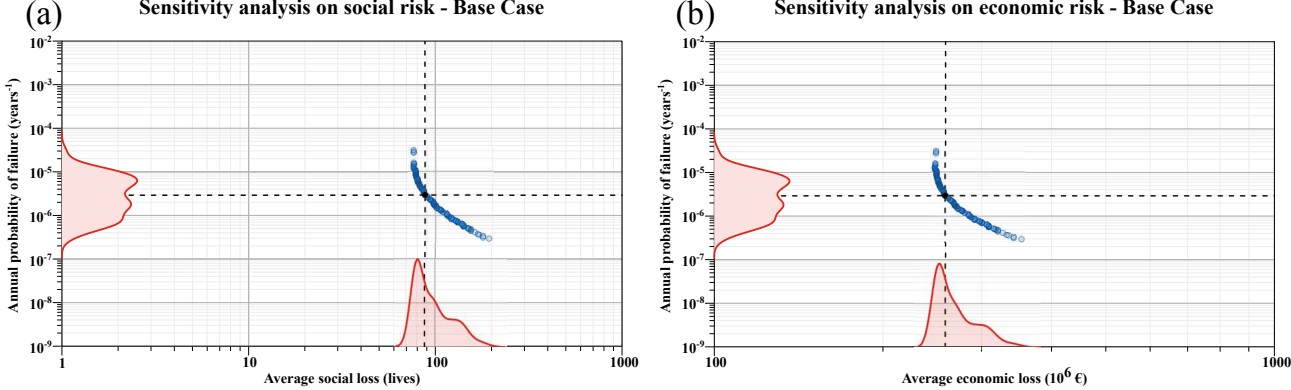

**Figure A2.** Effect of precipitation sampling uncertainty on (a) social risk, and (b) economic risk. The Kernel density plot for each variable is displayed in red on the x- and y-axes. The Base Case situation is highlighted with a black point and two dashed lines.