# Peer review of "Quantification of climate change impact on dam failure risk under hydrological scenarios: a case study from a Spanish dam"

_Natural Hazards and Earth System Sciences, 2019_

## Referee Comment (RC1) · Charles Rougé (Referee) · 14 Jun 2019

In this paper, authors propose a comprehensive framework for assessing dam failure risk under climate change, and apply their framework to a Spanish case. They use the classic definition of risk as the expected value of annual damages, use a sophisticated hydrological and water resource management model, consider several sources of failures and several climatic futures. Overall, this piece is a thorough and widely applicable methodology for engineers to assess risks to water infrastructure. It is of fairly good quality but could become excellent research with a thoughtful revision.

My main concerns are about 1) the lack of specifics and metrics for the hydrological

(and water resource) model validation, 2) the lack of critical analysis associated with the production of risk indicators associated to very high magnitude, very low probability events when only having a comparatively short flow record in hand, and 3) how results can be made meaningful for risk and resilience planning in a changing world.

I would advise authors to prioritise these aspects when preparing a revised version of this manuscript. A general remark is that authors seem keen to follow governmental guidelines and engineering practice, but the purpose of research should be to inform and improve these instead. For instance, research can do so by pointing out the limitations of engineering regulations and practice as a first step towards improving them. There is scope for authors, by engaging in a critical reflection of the assumptions they had to make, to shed light on the limitations of the current guidelines, and on which assumptions are critical. This would be a well-thought-out research publication with real-world consequences for engineering practice. This would require little extra modelling work, even though a basic sensitivity analysis would be a low-investment, high-reward endeavour in that it would highlight the parameters that have an outsized influence on the final risk indicators (and this would ease their discussion of which assumptions are critical).

1) Calibration / validation is not very precise on how much data there is to calibrate / validate against (how many years?), how the different periods are divided, or what the calibration criteria are. In particular, the quality of the fit (measured with Nash-Sutcliffe or Kling-Gupta) should be disclosed.

Figure 6: I understand this is simulated vs. observed daily flow from January 1st 2011 to December 31 2015. This should be made clearer, e.g. by inserting specific dates on the x-axis instead of having year labels at the middle of each year. Also on the same figure: why are the calibration results presented only for 2011-2015, a short five-year period? I understand the choice for figure presentation, but I would assume that the model is calibrated / validated against a longer period, and that should be made clear in the text.

More importantly maybe, it is not clear how the peak magnitudes in the model and observations match, and this is the crucial part of the flow for dam safety. Would it be possible to plot the differences between the two for days with flows have a certain threshold? Or for annual maxima (since those are the days used to derive the Gumbel distribution)? Likewise, the most important component of the hydrological model validation is whether the behaviour at high storage is reproduced for events during the historical record. In 2001 in particular, simulated reservoir levels are higher than observed levels: authors should understand why that is and what may be the consequences for their model-based risk assessment.

In summary for 1), authors should be more precise in the calibration and validation procedures, especially concerning the consequences of the choice of validation metrics on the risk assessment.

2) The lack of depth of the analysis of the relationships between modelling assumptions, the uncertainties associated with them, and modelling outcomes is not limited to the calibration of the hydrological and water resource models.

A central observation regarding this analysis is that risks to the dam are fully dependent on the existence of rare events whose probability of occurrence is extrapolated by fitting a predetermined distribution to short (< 100 data points) annual maxima time series. This has several consequences:

(i) The magnitudes of very long return period events will almost certainly be very sensitive to the parameters of the fitted distribution (not to mention the fact that other distributions than Gumbel's exist), so it seems compulsory to quantify the uncertainty on the Gumbel distribution's parameters, and understand the consequences of that uncertainty on the results.

(ii) Similarly, the choice of data (annual maxima vs. peak over threshold) may influence the estimates and therefore, risk indicators.

(iii) The formula linking hourly and daily intensity (equation (3)) is climate- and location-dependent, and the study explores future climates for which the parameters of this formula may change.

Authors could improve the value of their manuscript by showing how accounting for uncertainty in the estimates could change failure probabilities. Similarly, while readers can only appreciate the inclusion of gate performance indicators, the consequence of the assumptions of deterioration in performance on the overall results should be clarified (in other words, how does deterioration in gate performance affect future probabilities of failure?).

3) Maybe it is because the visuals are not conclusive (they often tend to show a risk increase but individual models / scenarios are all over the place), but it is not clear what the results mean for reservoir managers and planners. I would suggest for authors to present the ensemble of climate projections they use as... an ensemble, by deriving mean / uncertainty (e.g. standard deviation?) for each emission scenario and each period. Uncertainty analysis on the parameters that influence risk indicators calculation (see remarks (2)) would make these estimates of ensemble mean and standard deviation more robust and enable them to present their results in a way that can inform decision-making.

---

## Author Comment (AC1) · 28 Jun 2019

These are the Authors' replies to comments from Dr. Charles Rougé (Referee #1), received and published on 14 June 2019.

Firstly, we want to sincerely thank Referee #1 for the remarks and recommendations which will undoubtedly improve the quality and scope of the paper.

1) The main reason we haven't included a detailed description of the parameters driven the calibration/validation of the hydrological model is that we didn't want to overwhelm the reader with too much information. However, remarks of Referee #1 are very appro-

priate and will be taken into account when presenting the calibration in Section 5.2.1.

Figure 6 was displayed that way to be clearly readable, but also to present the hydrological data that was available for this article. For instance, the Barco de Ávila gauging station contains only 2 periods with valid data: - From 01/01/1971 to 01/07/1989. - From 01/10/2011 to 30/09/2015. Indeed, the x-axis in Figure 6 starts the 1st of October 2011 (first date of the 2011-2015 data period for the Barco de Ávila station) and ends the 30th of September 2015 (last date of hydrological data). This will be made clearer in the revised version.

The authors agree that a peak magnitudes analysis would benefit the clarity of the paper. The revised version will contain an evaluation of the relation between the discharge peak magnitudes in the model and in the observation series.

Results shown in Figure 7 do not correspond to the correct version of the model used. Instead, they correspond to a version where the seasonal maximum storage limitation was not yet implemented in the model and hence, they are invalid. The authors want to excuse for this mistake and will update a new figure with the correct series, in which the 2001 simulated levels do not exceed the observed ones.

2) (i) The authors agree with the remarks of Referee #1 regarding the uncertainty on the Gumbel distribution's parameters. A sensitivity analysis of the parameters would highlight how results are dependent on the pre-defined choices made. However, it is important to understand the computation cost with which we are dealing: a complete simulation, from the definition of the Gumbel distribution to the calculation of the dam risk, has an average duration of 24 hours. Thus, the computation duration of a sensitivity analysis (which entails several simulations for each case) applied to the ensemble of the 163 climatic models used would be incompatible with the publication deadlines imposed by the NHESS journal. However, the authors suggest performing this sensitivity analysis to the Base Case (present situation) and analyse the effect on its risk results; this would give an idea on how the other cases would react.

[Figure]

(ii) In this case, the annual maxima method has been arbitrary chosen among the different methods available. This has been selected since is a well know technique worldwide. Please refer to the previous author's comment for the convenience of applying a sensitivity analysis.

(iii) This is one of the main limitations of working with daily precipitation data: it is difficult to establish IDF relations when no sub-daily data is available. Thus, in order to deal with this issue, the option chosen was to rely on pre-defined formulations. Indubitably, the study could benefit from a more detailed analysis capable of producing a time-dependent relation for each climate model. However, this exceeds the scope of the paper. Nonetheless, a clarification of these issues and a justification of the method chosen will be included in the revised version of the paper.

Moreover, the effect on risk of gate performance deterioration is displayed in Figure 13. In this figure, the effect of each risk component (Previous pool level, Gate performance, Floods and Social consequences) has been isolated. It is however true that no clear explanation is presented in the text. This will be amended in the revised version of the paper.

3) We agree this is a key and complex aspect in the exploitation of such results. Although a certain general increase of the risk can be extracted from the results, it is difficult to directly define unequivocal recommendations for dam owners and managers. Different factors play important roles when assessing risk management action plans: Are risk acceptable in present situation? And in future scenarios? What are the risk reduction measures envisaged? How long should we wait until we implement them? What is the efficiency of each of these measures? What criteria should we follow to prioritize them? These are relevant questions that can be mentioned (but not resolved) in the paper. Thus, we will make sure that a more complete overview of the problem is introduced, which will help contextualize the usefulness of such approach. It is worth mentioning that this is a line of research that the authors are currently following: comprehensive decision-making support based on future changes in dam

risk. We invite Referee #1 and readers in general to track the authors' supplementary articles that explain next steps of the overall methodology and that are under review in other journals.

Please also note the supplement to this comment:
https://www.nat-hazards-earth-syst-sci-discuss.net/nhess-2019-141/nhess-2019-141-AC1-supplement.pdf

**Supplement:**

**AUTHOR'S RESPONSES TO REFEREE #1**

These are the Authors' replies to comments from Dr. Charles Rougé (Referee #1), received and published on 14 June 2019. We use blue colour for our replies and black colour for Referee's comments.

**RESPONSES**:

Firstly, we want to sincerely thank Referee #1 for the remarks and recommendations which will undoubtedly improve the quality and scope of the paper.

My main concerns are about 1) the lack of specifics and metrics for the hydrological (and water resource) model validation, 2) the lack of critical analysis associated with the production of risk indicators associated to very high magnitude, very low probability events when only having a comparatively short flow record in hand, and 3) how results can be made meaningful for risk and resilience planning in a changing world.

I would advise authors to prioritise these aspects when preparing a revised version of this manuscript. A general remark is that authors seem keen to follow governmental guidelines and engineering practice, but the purpose of research should be to inform and improve these instead. For instance, research can do so by pointing out the limitations of engineering regulations and practice as a first step towards improving them. There is scope for authors, by engaging in a critical reflection of the assumptions they had to make, to shed light on the limitations of the current guidelines, and on which assumptions are critical. This would be a well-thought-out research publication with real world consequences for engineering practice. This would require little extra modelling work, even though a basic sensitivity analysis would be a low-investment, high-reward endeavour in that it would highlight the parameters that have an outsized influence on the final risk indicators (and this would ease their discussion of which assumptions are critical).

We will do our best to address the issues raised by Referee #1.

1) Calibration / validation is not very precise on how much data there is to calibrate / validate against (how many years?), how the different periods are divided, or what the calibration criteria are. In particular, the quality of the fit (measured with Nash-Sutcliffe or Kling-Gupta) should be disclosed.

The main reason we haven't included a detailed description of the parameters driven the calibration/validation of the hydrological model is that we didn't want to overwhelm the reader with too much information. However, remarks of Referee #1 are very appropriate and will be taken into account when presenting the calibration in Section 5.2.1.

Figure 6: I understand this is simulated vs. observed daily flow from January 1st 2011 to December 31 2015. This should be made clearer, e.g. by inserting specific dates on the x-axis instead of having year labels at the middle of each year. Also on the same figure: why are the calibration results presented only for 2011-2015, a short five-year period? I understand the choice for figure presentation, but I would assume that the model is calibrated / validated against a longer period, and that should be made clear in the text.

Figure 6 was displayed that way to be clearly readable, but also to present the hydrological data that was available for this article. For instance, the Barco de Ávila gauging station contains only 2 periods with valid data:

- From 01/01/1971 to 01/07/1989.
- From 01/10/2011 to 30/09/2015.
Indeed, the x-axis in Figure 6 starts the 1st of October 2011 (first date of the 2011-2015 data period for the Barco de Ávila station) and ends the 30th of September 2015 (last date of hydrological data). This will be made clearer in the revised version.

More importantly maybe, it is not clear how the peak magnitudes in the model and observations match, and this is the crucial part of the flow for dam safety. Would it be possible to plot the differences between the two for days with flows have a certain threshold? Or for annual maxima (since those are the days used to derive the Gumbel distribution)?

The authors agree that this analysis would benefit the clarity of the paper. The revised version will contain an evaluation of the relation between the discharge peak magnitudes in the model and in the observation series.

Likewise, the most important component of the hydrological model validation is whether the behaviour at high storage is reproduced for events during the historical record. In 2001 in particular, simulated reservoir levels are higher than observed levels: authors should understand why that is and what may be the consequences for their model-based risk assessment.

Results shown in Figure 7 do not correspond to the correct version of the model used. Instead, they correspond to a version where the seasonal maximum storage limitation was not yet implemented in the model and hence, they are invalid. The authors want to excuse for this mistake and will update a new figure with the correct series, in which the 2001 simulated levels do not exceed the observed ones.

2) The lack of depth of the analysis of the relationships between modelling assumptions, the uncertainties associated with them, and modelling outcomes is not limited to the calibration of the hydrological and water resource models.

A central observation regarding this analysis is that risks to the dam are fully dependent on the existence of rare events whose probability of occurrence is extrapolated by fitting a predetermined distribution to short (< 100 data points) annual maxima time series. This has several consequences:

(i) The magnitudes of very long return period events will almost certainly be very sensitive to the parameters of the fitted distribution (not to mention the fact that other distributions than Gumbel's exist), so it seems compulsory to quantify the uncertainty on the Gumbel distribution's parameters, and understand the consequences of that uncertainty on the results.

The authors agree with the remarks of Referee #1. A sensitivity analysis of the parameters would highlight how results are dependent on the pre-defined choices made. However, it is important to understand the computation cost with which we are dealing: a complete simulation, from the definition of the Gumbel distribution to the calculation of the dam risk, has an average duration of 24 hours. Thus, the computation duration of a sensitivity analysis (which entails several simulations for each case) applied to the ensemble of the 163 climatic models used would be incompatible with the publication deadlines imposed by the NHESS journal. However, the authors suggest performing this sensitivity analysis to the Base Case (present situation) and analyse the effect on its risk results; this would give an idea on how the other cases would react.

(ii) Similarly, the choice of data (annual maxima vs. peak over threshold) may influence the estimates and therefore, risk indicators.

In this case, the annual maxima method has been arbitrary chosen among the different methods available. This has been selected since is a well know technique worldwide. Please refer to the previous author's comment for the convenience of applying a sensitivity analysis.

(iii) The formula linking hourly and daily intensity (equation (3)) is climate- and location dependent, and the study explores future climates for which the parameters of this formula may change.

This is one of the main limitations of working with daily precipitation data: it is difficult to establish IDF relations when no sub-daily data is available. Thus, in order to deal with this issue, the option chosen was to rely on pre-defined formulations.

Indubitably, the study could benefit from a more detailed analysis capable of producing a time-dependent relation for each climate model. However, this exceeds the scope of the paper.

Nonetheless, a clarification of these issues and a justification of the method chosen will be included in the revised version of the paper.

Authors could improve the value of their manuscript by showing how accounting for uncertainty in the estimates could change failure probabilities. Similarly, while readers can only appreciate the inclusion of gate performance indicators, the consequence of the assumptions of deterioration in performance on the overall results should be clarified (in other words, how does deterioration in gate performance affect future probabilities of failure?).

The effect on risk of gate performance deterioration is displayed in Figure 13. In this figure, the effect of each risk component (Previous pool level, Gate performance, Floods and Social consequences) has been isolated. It is however true that no clear explanation is presented in the text. This will be amended in the revised version of the paper.

3) Maybe it is because the visuals are not conclusive (they often tend to show a risk increases but individual models / scenarios are all over the place), but it is not clear what the results mean for reservoir managers and planners. I would suggest for authors to present the ensemble of climate projections they use as... an ensemble, by deriving mean / uncertainty (e.g. standard deviation?) for each emission scenario and each period. Uncertainty analysis on the parameters that influence risk indicators calculation (see remarks (2)) would make these estimates of ensemble mean and standard deviation more robust and enable them to present their results in a way that can inform decision-making.

We agree this is a key and complex aspect in the exploitation of such results. Although a certain general increase of the risk can be extracted from the results, it is difficult to directly define unequivocal recommendations for dam owners and managers. Different factors play important roles when assessing risk management action plans: Are risk acceptable in present situation? And in future scenarios? What are the risk reduction measures envisaged? How long should we wait until we implement them? What is the efficiency of each of these measures? What criteria should we follow to prioritize them? These are relevant questions that can be mentioned (but not resolved) in the paper. Thus, we will make sure that a more complete overview of the problem is introduced, which will help contextualize the usefulness of such approach.

It is worth mentioning that this is a line of research that the authors are currently following: comprehensive decision-making support based on future changes in dam risk. We invite Referee #1 and readers in general to track the authors' supplementary articles that explain next steps of the overall methodology and that are under review in other journals.

---

## Referee Comment (RC2) · Anonymous Referee #2 · 5 Aug 2019

The manuscript contributes to the knowledge of how climate change might affect dam risk management and the definition of long-term strategies to reduce risk. The paper presents a method for addressing different climate change scenarios to evaluate their influence on future risk. The results obtained for the case study show how dam failure risk may vary depending on different scenarios and identify the most influencing factors regarding affected risk components (e.g. estimated income floods or reservoir levels). The proposed method can be applied to evaluate the potential impact of climate change in other cases and provides a tool for applying a dynamic approach in terms of risk analysis and management. This topic fits well into the scope of the Journal. In my opinion, the methodological approach and the discussion of results are of interest

for the general audience of the journal and the paper deserves publication. However, I include several specific questions which clarification could improve the quality and understandability of the paper, and should be addressed: • Page 3. Line 15. Regarding the definition of study periods, for the Base Case, please further explain the reasons why the period 1970-2005 is selected and whether it is proposed as a general approach. • Section 3 would require further description on why this case study has been chosen for the conducted research and why the analysis of the impact of climate change is of interest for this dam. • Section 4. Page 9. Please further describe why different maximum water pool levels per month are considered for the case study. • Section 5.2.1. A more detailed description of the calibration process for the hydrologic-hydraulic model is required (parameters calibrated, efficiency indicators used, etc.). • Section 5.3.1. Page 17. Line 20. The authors introduce the concept of event tree not yet described up to this point. Please contextualize the link between the proposed risk model and the event tree mentioned in this section. • Section 5.3.4. Further details on how variations on the population and water supply demands are considered in future scenarios in terms of potential economic consequences (i.e. in terms of future demands) would be convenient. Do the authors consider that provided services remain unchanged in future scenarios? • Section 7. Conclusions: o The added value of using risk models to integrate information on projected effects of climate change is highlighted, however, how the proposed approach can be adapted to low-data available cases? o In terms of supporting dam safety management, how results for this case study will influence long-term actions for this dam? Please describe how obtained results can be considered for the definition of future actions (for instance, in terms of new operating rules or water pool levels). o A sensitivity analysis have been included to evaluate the impact on risk of each factor independently. A short discussion regarding uncertainty analysis would improve this section (e.g. their influence on risk outcomes). In addition, please note the following suggestions regarding technical corrections: • The size, quality and readability of figures is very good in general, although some figures might be improved (e.g. Figure 10). • A list of minor

corrections is here included:

Page Line Comment 1 29 these impacts 5 Figure 1 assess climate change impacts on 5 6 dam built in 1960 5 9 concrete gravity dam 6 12 AEMET. Please describe acronym. 6 17 CEDEX. Please describe acronym. 6 18 in this same platform. 8 15 of potential incoming floods 8 16 from the flood routing 8 17 gate availability 9 6 consequences for the non-failure case 9 26 reservoir's releases 9 29 is the limited water storage in the Santa Teresa reservoir 9 31 Consider replacing "in the computation of the risk model since they define the maximum possible water level issued from the study of previous pool levels" by "for estimating water pool levels" 12 11 divided into subbasins 12 17 The calibration process presents 13 13 Check sentence construction 13 15 Concerning basin discharges 14 5 by the Hydrological Plan 14 8 Consider replacing "The validation of this water resources model" by "model validation" 14 10 As shown 14 10 results performance 14 14 is not capable 14 16 Once the model is validated 15 21 Consider replacing "The time distribution of the rainfall" by "Temporal rainfall distribution" 17 1 Consider replacing "The characterisation of the peak discharges with their return Period" by "Peak discharge by return period" 17 11 and the social consequences used to compute the social risk 17 12 other risk model components 17 11 spillway gate and bottom outlet performance 17 15 Increasingly 19 7 Consider replacing "will go from" by "will vary" 20 19 Reference is required 25 Tables RCP2.6

---

## Author Comment (AC2) · 10 Aug 2019

These are the Authors' replies to comments from Referee #2, received and published on 5 August 2019. Firstly, we want to sincerely thank Referee #2 for the remarks and recommendations which will undoubtedly improve the quality and scope of the paper.

- The 1970-2005 period has been selected as the Base Case because it was the longest period for which we had both observed and historical data (for the climate projections).

- A risk analysis was already applied to the Santa Teresa dam in a previous study

(Ardiles et al., 2011; Morales-Torres et al., 2016). Results from this study showed that, although the dam didn't required urgent correction measures, its risk was important enough to be carefully monitored. Thus, we considered interesting to evaluate if the risk situation of the dam was expecting to increase and thus immediate actions were necessary, or if its risk was expected to decrease until no urgent actions were necessary.

- In this particular case, different maximum water pool levels are considered for each month because of the expected seasonality of high flows which tend to increase in winter (December to February). In prevision of important water volumes entering the reservoir, the dam exploiters increase the freeboard volume to absorb them. These exploitation rules are contained in the Hydrological Plan of the Duero River Basin (Confederación Hidrográfica del Duero, 2015).

- Please refer to the author's response to Referee #1 concerning this matter. More details about the calibration process will be included in the reviewed version of the paper.

- Event trees help representing all the possible chains of events resulting from an initiating event and are used as a basis for the dam risk model used in the manuscript. A detailed description will be included in the reviewed version of the paper.

- Although population and water demands are supposed variable with time in the paper, for simplicity no new services are considered in the future. This will be included in the reviewed version of the paper.

- Conclusions:

* The paradigm of low-data study cases has not been considered in this work. Under such circumstances, another approach might be of use. We encourage Referee #2 to consult a previous paper of the authors (Fluixá-Sanmartín et al., 2018) where this situation is tackled.

[Figure]

* Although this work represents a useful tool for dam safety management, it is clear that further analyses are required before decision can be made. In particular, the uncertainty associated to future risks imposes a deeper evaluation of the recommendations to make. However, this problem as well as some suggestions will be mentioned in the new version of the paper.

* As suggested by Referee #2, a discussion on uncertainty will be included in the conclusions section.

- The composition of the figures will be re-evaluated to increase their readability.

- List of minor corrections: we will take into account the Referee's remarks and corrections and will include them in the new version of the paper.

REFERENCES

Ardiles, L., Sanz, D., Moreno, P., Jenaro, E., Fleitz, J. and Escuder-Bueno, I.: Risk Assessment and Management for 26 Dams Operated By the Duero River Authority (Spain), in 6th International Conference on Dam Engineering, C.Pina, E.Portela, J.Gomes, Lisbon, Portugal., 2011.

Confederación Hidrográfica del Duero: Plan Hidrológico de la parte española de la demarcación hidrográfica del Duero. 2015-2021. [online] Available from: www.chduero.es, 2015.

Fluixá-Sanmartín, J., Altarejos-García, L., Morales-Torres, A. and Escuder-Bueno, I.: Review article: Climate change impacts on dam safety, Natural Hazards and Earth System Sciences, 18(9), 2471–2488, doi:10.5194/nhess-18-2471-2018, 2018.

Morales-Torres, A., Serrano-Lombillo, A., Escuder-Bueno, I. and Altarejos-García, L.: The suitability of risk reduction indicators to inform dam safety management, Structure and Infrastructure Engineering, 1–12, doi:10.1080/15732479.2015.1136830, 2016.

Please also note the supplement to this comment:

[Figure]

https://www.nat-hazards-earth-syst-sci-discuss.net/nhess-2019-141/nhess-2019-141-AC2-supplement.pdf

**Supplement:**

**AUTHOR'S RESPONSES TO REFEREE #2**

These are the Authors' replies to comments from Referee #2, received and published on 5 August 2019. We use blue colour for our replies and black colour for Referee's comments.

**RESPONSES**:

Firstly, we want to sincerely thank Referee #2 for the remarks and recommendations which will undoubtedly improve the quality and scope of the paper.

The manuscript contributes to the knowledge of how climate change might affect dam risk management and the definition of long-term strategies to reduce risk. The paper presents a method for addressing different climate change scenarios to evaluate their influence on future risk. The results obtained for the case study show how dam failure risk may vary depending on different scenarios and identify the most influencing factors regarding affected risk components (e.g. estimated income floods or reservoir levels). The proposed method can be applied to evaluate the potential impact of climate change in other cases and provides a tool for applying a dynamic approach in terms of risk analysis and management. This topic fits well into the scope of the Journal. In my opinion, the methodological approach and the discussion of results are of interest for the general audience of the journal and the paper deserves publication.

However, I include several specific questions which clarification could improve the quality and understandability of the paper, and should be addressed:

- Page 3. Line 15. Regarding the definition of study periods, for the Base Case, please further explain the reasons why the period 1970-2005 is selected and whether it is proposed as a general approach.
  The 1970-2005 period has been selected as the Base Case because it was the longest period for which we had both observed and historical data (for the climate projections).
- Section 3 would require further description on why this case study has been chosen for the conducted research and why the analysis of the impact of climate change is of interest for this dam.
  A risk analysis was already applied to the Santa Teresa dam in a previous study (Ardiles et al., 2011; Morales-Torres et al., 2016). Results from this study showed that, although the dam didn't required urgent correction measures, its risk was important enough to be carefully monitored. Thus, we considered interesting to evaluate if the risk situation of the dam was expecting to increase and thus immediate actions were necessary, or if its risk was expected to decrease until no urgent actions were necessary.
- Section 4. Page 9. Please further describe why different maximum water pool levels per month are considered for the case study.
  In this particular case, different maximum water pool levels are considered for each month because of the expected seasonality of high flows which tend to increase in winter (December to February). In prevision of important water volumes entering the reservoir, the dam exploiters increase the freeboard volume to absorb them. These exploitation rules are contained in the Hydrological Plan of the Duero River Basin (Confederación Hidrográfica del Duero, 2015).
- Section 5.2.1. A more detailed description of the calibration process for the hydrologic-hydraulic model is required (parameters calibrated, efficiency indicators used, etc.).
  Please refer to the author's response to Referee #1 concerning this matter. More details about the calibration process will be included in the reviewed version of the paper.

- Section 5.3.1. Page 17. Line 20. The authors introduce the concept of event tree not yet described up to this point. Please contextualize the link between the proposed risk model and the event tree mentioned in this section.
  Event trees help representing all the possible chains of events resulting from an initiating event and are used as a basis for the dam risk model used in the manuscript. A detailed description will be included in the reviewed version of the paper.
- Section 5.3.4. Further details on how variations on the population and water supply demands are considered in future scenarios in terms of potential economic consequences (i.e. in terms of future demands) would be convenient. Do the authors consider that provided services remain unchanged in future scenarios?
  Although population and water demands are supposed variable with time in the paper, for simplicity no new services are considered in the future. This will be included in the reviewed version of the paper.
- Section 7. Conclusions:
  - The added value of using risk models to integrate information on projected effects of climate change is highlighted, however, how the proposed approach can be adapted to low-data available cases?
    The paradigm of low-data study cases has not been considered in this work. Under such circumstances, another approach might be of use. We encourage Referee #2 to consult a previous paper of the authors (Fluixá-Sanmartín et al., 2018) where this situation is tackled.
  - In terms of supporting dam safety management, how results for this case study will influence long-term actions for this dam? Please describe how obtained results can be considered for the definition of future actions (for instance, in terms of new operating rules or water pool levels).
    Although this work represents a useful tool for dam safety management, it is clear that further analyses are required before decision can be made. In particular, the uncertainty associated to future risks imposes a deeper evaluation of the recommendations to make. However, this problem as well as some suggestions will be mentioned in the new version of the paper.
  - A sensitivity analysis has been included to evaluate the impact on risk of each factor independently. A short discussion regarding uncertainty analysis would improve this section (e.g. their influence on risk outcomes).
    As suggested by Referee #2, a discussion on uncertainty will be included in the conclusions section.

In addition, please note the following suggestions regarding technical corrections:

- The size, quality and readability of figures is very good in general, although some figures might be improved (e.g. Figure 10).
  The composition of the figures will be re-evaluated to increase their readability.
- A list of minor corrections is here included:

| Page | Line | Comment |
| --- | --- | --- |
| 1 | 29 | these impacts |
| 5 | Figure 1 | assess climate change impacts on |
| 5 | 6 | dam built in 1960 |
| 5 | 9 | concrete gravity dam |
| 6 | 12 | AEMET. Please describe acronym. |
| 6 | 17 | CEDEX. Please describe acronym. |

| 6 | 18 | in this same platform. |
|---|---|---|
| 8 | 15 | of potential incoming floods |
| 8 | 16 | from the flood routing |
| 8 | 17 | gate availability |
| 9 | 6 | consequences for the non-failure case |
| 9 | 26 | reservoir's releases |
| 9 | 29 | is the limited water storage in the Santa Teresa reservoir |
| 9 | 31 | Consider replacing "in the computation of the risk model since they define the maximum possible water level issued from the study of previous pool levels" by "for estimating water pool levels" |
| 12 | 11 | divided into subbasins |
| 12 | 17 | The calibration process presents |
| 13 | 13 | Check sentence construction |
| 13 | 15 | Concerning basin discharges |
| 14 | 5 | by the Hydrological Plan |
| 14 | 8 | Consider replacing "The validation of this water resources model" by "model validation" |
| 14 | 10 | As shown |
| 14 | 10 | results performance |
| 14 | 14 | is not capable |
| 14 | 16 | Once the model is validated |
| 15 | 21 | Consider replacing "The time distribution of the rainfall" by "Temporal rainfall distribution" |
| 17 | 1 | Consider replacing "The characterisation of the peak discharges with their return Period" by "Peak discharge by return period" |
| 17 | 11 | and the social consequences used to compute the social risk |
| 17 | 12 | other risk model components |
| 17 | 11 | spillway gate and bottom outlet performance |
| 17 | 15 | increasingly |
| 19 | 7 | Consider replacing "will go from" by "will vary" |
| 20 | 19 | Reference is required |
| 25 | Tables | RCP2.6 |

We will take into account the Referee's remarks and corrections and will include them in the new version of the paper.

**REFERENCES**

Ardiles, L., Sanz, D., Moreno, P., Jenaro, E., Fleitz, J. and Escuder-Bueno, I.: Risk Assessment and Management for 26 Dams Operated By the Duero River Authority (Spain), in 6th International Conference on Dam Engineering, C.Pina, E.Portela, J.Gomes, Lisbon, Portugal., 2011.

Confederación Hidrográfica del Duero: Plan Hidrológico de la parte española de la demarcación hidrográfica del Duero. 2015-2021. [online] Available from: www.chduero.es, 2015.

Fluixá-Sanmartín, J., Altarejos-García, L., Morales-Torres, A. and Escuder-Bueno, I.: Review article: Climate change impacts on dam safety, Natural Hazards and Earth System Sciences, 18(9), 2471–2488, doi:10.5194/nhess-18-2471-2018, 2018.

Morales-Torres, A., Serrano-Lombillo, A., Escuder-Bueno, I. and Altarejos-García, L.: The suitability of risk reduction indicators to inform dam safety management, Structure and Infrastructure Engineering, 1–12, doi:10.1080/15732479.2015.1136830, 2016.